

# Scenario building and runout modelling for debris flow hazards in pro-/periglacial catchments with scarce past event data: application of a multi-methods approach for the Dar catchment (western Swiss Alps)

Mauro Fischer[1,2], Mario Kummert[1], Reto Aeschbacher[1,3], Christoph Graf[4], Alexis Rüeger[1], Philippe Schoeneich[5], Markus Zimmermann[1,2,6], and Margreth Keiler[1,2,7,8]

[1]Institute of Geography, University of Bern, Bern, Switzerland
[2]Oeschger Centre for Climate Change Research, University of Bern, Bern, Switzerland
[3]Kissling + Zbinden AG, Bern, Switzerland
[4]Swiss Federal Institute for Forest, Snow and Landscape Research WSL, Birmensdorf, Switzerland
[5]Institute of Urbanism and Alpine Geography, University of Grenoble Alps, Grenoble, France
[6]NDR Consulting GmbH, Thun, Switzerland
[7]Department of Geography, University of Innsbruck, Innsbruck, Austria
[8]Institute for Interdisciplinary Mountain Research, Austrian Academy of Sciences, Innsbruck, Austria

**Correspondence:** Mauro Fischer (mauro.fischer@unibe.ch)

**Abstract.** In high mountain areas, the disposition (susceptibility of occurrence) for debris flows is increasing in steep terrain, as – due to climate change – rapid glacier retreat and permafrost degradation is favouring higher availability of loose sediments. The probability of occurrence and magnitude of pro- and periglacial debris flows is increasing, too, as triggering events such as heavy thunderstorms, long-lasting rainfalls, intense snow melt or rain-on-snow events are likely to occur more often and more

intensely in future decades. Hazard assessment for debris flows originating from pro- and periglacial areas is thus crucial but remains challenging, as records of past events on which local magnitude-frequency relationships and debris flow scenarios can be based on are often scarce or inexistent. In this study, we present a multi-methods approach for debris flow hazard scenario building and runout modelling in pro- and periglacial catchments with scarce past event data. Scenario building for the debris flow initiation zone reposes on (i) the definition of meteorological and hydrological triggering scenarios using data on extreme

point rainfall and precipitation-runoff modelling, and (ii) the definition of bed load scenarios from empirical approaches and field surveys. Numerical runout modelling and hazard assessment for the resulting debris flow scenarios is carried out using RAMMS-DF, which was calibrated to the studied catchment (Le Dar, western Swiss Alps) based on the area of debris flow deposits from the single major event recorded there in summer 2005. The developed approach is among the first to propose systematic scenario building for pro- and periglacial debris flows triggered by precipitation dependent events.

# 1   Introduction

Debris flows are fast-flowing mass movements behaving like a fluid and consisting of a mixture of water and (typically 30 to 60%) fine to coarse-grained sediments (Iverson et al., 1997). In the initiation area, the triggering of debris flows requires



steep slope gradients (>20%; Rickenmann, 1997; Jakob and Hungr, 2005), unconsolidated fine- and coarse-grained sediments, a certain amount of water, and sparse or absent vegetation (Costa, 1984). Due to their potential to transport important masses

of destructive coarse debris at high velocity and over long runout distances, debris flows represent one of the most important types of natural hazards in the Alps (Rickenmann, 1997). Their relative unpredictability reinforces their hazardous character (Jakob and Hungr, 2005). The occurrence, magnitude and consequences of debris flows are strongly influenced by ongoing climate and environmental changes as well as human activities (Keiler et al., 2010; Stoffel et al., 2014).

Especially in (cold) high mountain areas several challenges exist related to future debris flow hazard and risk assessment.

At high elevation (above the tree line), steep and non-vegetated sediment sources, highly susceptible to the triggering of debris flows, are largely found in pro- and periglacial areas (Zimmermann and Haeberli, 1992). Proglacial areas (or glacier forefields) designate the area between the Little Ice Age (LIA) termino-lateral moraines and contemporary glacier extents (Heckmann and Morche, 2019). The term periglacial refers to zones dominated by processes related to seasonal frost and/or permafrost (French, 2017). Here, pro-/periglacial debris flows (hereinafter abbreviated as PPDFs) are defined as debris flows starting within glacier

forefields or areas outside the LIA glacier extents with seasonal to perennial frost.

Due to climate change, both pro- and periglacial environments are subject to rapid and irreversible changes (e.g., Beniston et al., 2018). Indeed, mountain glaciers have shown particularly strong mass loss and shrinkage over the last decades (Zemp et al., 2015, 2019; Paul et al., 2020; Hugonnet et al., 2021), and continued global warming will most likely cause the disappearance of a vast majority of alpine glaciers till the end of the 21st century (Zekollari et al., 2019). In consequence, proglacial areas

are continuously enlarging (Carrivick et al., 2018), meaning that the amount of loose material (glacigenic sediments or loose material of other origin) susceptible to remobilisation and transport by gravitational or fluvial processes such as debris flows is steadily increasing over time (Ballantyne, 2002; Carrivick and Heckmann, 2017). In periglacial areas, climate change is known to limit ground surface freezing and to cause both permafrost warming and active layer thickening (Biskaborn et al., 2019; PERMOS, 2022). As a result, increasing slope instabilities and thus increasing sediment release through rock fall (e.g., Gruber

et al., 2004; Ravanel et al., 1997), rock faces destabilization (e.g., Walter et al., 2020; Hendrickx et al., 2022) and rock glaciers acceleration and destabilization (e.g., PERMOS, 2019; Marcer et al., 2021; Kummert et al., 2018, 2021) have been observed and documented over the past twenty years, and are likely to continue occurring over the next decades.

Hence, in both high mountain pro- and periglacial environments, the availability of unconsolidated sediments is increasing. The disposition (susceptibility of occurrence, cf. Zimmermann et al., 1997a) and potential magnitude (volume per event, cf. Jakob and Friele, 2010) of PPDFs is thus expected to increase in areas with slopes potentially steep enough for debris flow

initiation (e.g., Stoffel and Graf, 2015). PPDFs can be triggered by (i) precipitation dependent events (thunderstorms, long-lasting rainfalls, rain-on-snow (ROS) events) or (ii) precipitation independent events (snow melt, glacier lake or englacial water pocket outburst floods (GLOFs, WPOFs)) (e.g., Wieczorek and Glade, 2005; Decaulne and Sæmundsson, 2007; Guzzetti et al., 2008). With climate change, both are likely to become more frequent and more intense (Lenderink and Van Meijgaard,

2008; Floris et al., 2010; Gobiet et al., 2014; Hirschberg et al., 2021). Thus, the probability of occurrence (or frequency) of PPDFs is expected to increase, too. In this context, building PPDF scenarios that consider the above-mentioned changes to high mountain areas is essential for future hazard and risk management strategies (Hirschberg et al., 2021).



In general, methodological approaches aiming to assess debris flow hazards are based on, or validated against, data from past events (e.g., Huggel et al., 2004; Hürlimann et al., 2006; Rickenmann et al., 2006; Jakob et al., 2013). Records from past debris flows (e.g., event cadastres and practitioners' reports) or historical event data as for instance derived by dendrochronological methods are often used to build a torrent-specific magnitude-frequency (M-F) relation describing debris flow occurrence on alluvial fans (e.g., Zimmermann et al., 1997a; Jakob and Friele, 2010; Stoffel, 2010). Past events are classified by size and assigned a return period. Hazard mapping and mitigation measures are then developed based on both the intensity and probability of occurrence of the events, which are directly derived from the M-F relationship (Raetzo et al., 2002).

For highly dynamic and currently imbalanced systems such as high mountain pro- and periglacial areas, it remains however questionable to assess future debris flow hazards based on the analysis of past events (e.g., Schneider et al., 2014; Heiser et al., 2022). In addition, records of past debris flows are in many cases inexistent or incomplete (Chiarle et al., 2007; Frey et al., 2016). For torrents with scarce or no data on past events, potential future debris flow hazards are generally assessed through approximation to similar cases or – even more generalized – through the application of empirical relationships (e.g., Rickenmann, 1999) that do not allow to take into account the local specificities of each catchment. To our knowledge, very few studies have so far come up with systematic scenario building approaches for pro- and periglacial debris flows (cf. Allen et al., 2022). In addition, while it is known that PPDFs are actually also often triggered by heavy precipitation events or/in combination with intense snow melt (Rickenmann and Zimmermann, 1993; Chiarle et al., 2007), the few illustrative examples of existing PPDF scenario building approaches mainly focus on debris flows triggered by GLOFs (e.g., Schneider et al., 2014; Frey et al., 2018). There is thus a need for methods that allow to build precipitation-related debris flow scenarios without having to resort to comprehensive past events documentation and that do not repose exclusively on empirical relationships.

In this study, we propose a multi-methods approach to elaborate scenarios for PPDFs and thus to assess debris flow hazard potential in catchments with scarce records of past events. The developed approach reposes on (i) the definition of precipitation dependent debris flow triggering scenarios and the calculation of associated runoff, (ii) the estimation of bed load volumes potentially triggered per scenario, (iii) the assignation of a potential total event volume (water-debris-mixture) to each triggering scenario and (iv) the definition of debris flow (sub-)scenarios for individual total event volumes (event sequence, number of surges, hydrograph per surge, flow characteristics, potential triggering conditions). The resulting debris flow scenarios then serve as input for numerical debris flow modelling (here with RAMMS-DF, cf. Bartelt et al., 2017) to assess flow paths, flow heights, velocities, runout distances, as well as transported and deposited sediment volumes of future potential PPDFs. We apply the proposed approach to the case study of the Dar torrential catchment located in the western Swiss Alps. On a comparatively small scale, this catchment represents a very good example for the current rapid and dynamic changes in the high mountain cryosphere and the challenges related to the hazard assessment of future PPDFs in catchments with scarce event data.





**Figure 1.** The Dar catchment. (a) Location and general overview. Underlain is a 2013 SWISSIMAGE orthophoto from swisstopo. The light blue triangle indicates the defined location of the input hydrograph for numerical debris flow modelling with RAMMS-DF (chapter 4.1.4). (b) Cutout of the geotechnical map of Switzerland (1:200'000) for the entire catchment (De Quervain et al., 1965). (c) Longitudinal profile from the highest to the lowest point of the catchment (from the source of the Dar torrent below the first rock step until the confluence with La Grande Eau always along the torrent) based on high-resolution ALS-DEMs acquired between 2001 and 2005 by the Canton of Vaud (Office de l'information sur le Territoire, 2002). (d) Runoff regime of the Dar torrent based on modelled data (FOEN, 2000).

## 2 The Dar catchment

### 2.1 General characteristics and overview

The Dar torrent in the municipality of Ormont-Dessus (Canton of Vaud, western Swiss Alps) has its source at the newly emerging proglacial lake of Glacier du Sex Rouge at ca. 2745 m a.s.l. (Fig. A1 and Fig. B1d). Its catchment extends over an area of 10.53 km² and ranges in elevation between 3123 m a.s.l. (Oldehore/Becca d'Audon) and 1174 m a.s.l. at the confluence



with La Grande Eau near the village of Les Diablerets (Fig. 1a,c). The torrent initially flows about 2.5 km in north-northwest

direction in steep and fairly open terrain, crossing three striking steep rock steps with waterfalls (Fig. 1a,c). After the third rock step (Cascade du Dar), it shows a sharp left turn and abrupt change in slope. From there (Creux du Pillon), it follows a much more gently inclined valley floor westwards for another 2.8 km until the village of Les Diablerets (Fig. 1a,c, Fig. A1).

The climate of the Dar catchment is cold-temperate, with a mean annual air temperature around 2.5°C and an average annual precipitation >1600 mm (values for the climate normal period 1991–2020 upscaled to the mean catchment elevation

(1892 m a.s.l.) from nearby MeteoSwiss weather stations). The Dar shows a typical nivo-glacial runoff regime with a high runoff variability throughout the year (modelled monthly averages between roughly 0.1 m³/s in January and 1.0 m³/s in June) (FOEN, 2000, Fig. 1d). Maximum runoff occurs from May to August, with a clear peak in June, which points to the relatively higher importance of snow melt to runoff generation in the catchment compared to ice melt.

Different geology/lithologies clearly shape today's predominant relief and landforms in the Dar catchment (De Quervain et

al., 1965; Badoux and Gabus, 1991; Schoeneich and Reynard, 2021, Fig. 1b). The steep slopes and higher mountains south of the Les Diablerets-Col du Pillon axis are mainly composed of different limestones. In the central part, parallel to the valley floor, the substratum is made of late Pleistocene moraine material (glacial till) or gypsum and anhydrites. In the northernmost part of the Dar catchment, with more gentle slopes and lower peaks and ridges, flysch, glacial till and other Quaternary sediments are dominant (Fig. 1a,b).

**2.2  The cirque of Le Dar Dessus – initiation zone for PPDFs**

Forming the initiation zone for pro- and periglacial debris flows in the Dar catchment, the cirque of Le Dar Dessus between the first and second steep rock step (Fig. 1a,c, Fig. A1) consists of extensive sediment deposits of either glacial, gravitational or fluvially reworked origin ("mixed accumulation area" in Fig. 2a, cf. Lambiel et al., 2008). In general, the uppermost parts of the cirque are dominated by gravitational processes, whereas a transition between gravitational and fluvial processes can be

observed in the middle sections, followed by fluvial deposits towards the lower boundaries of the cirque, just upstream from the second steep rock step (Lambiel et al., 2008). At ca. 2280 m a.s.l., parts of the LIA terminal moraine of Glacier du Sex Rouge are still clearly recognizable (Fig. 2a, Fig. A1b).

Remarkable shrinkage and mass loss has been observed for glaciers in the Dar catchment and adjacent glaciers of the entire Diablerets massif during recent decades (Fischer et al., 2015, 2016; Fischer, 2018; GLAMOS, 2021; Linsbauer et al., 2021).

At the end of the Little Ice Age, Glacier du Sex Rouge and Glacier du Dar were connected and formed one single glacier of 0.93 km² (cf. Swiss Glacier Inventory SGI1850, Maisch et al., 2000; Paul, 2004), covering large parts of the cirque of Le Dar Dessus (Fig. 2b). Today, only 2.75% of the surface area of the Dar catchment is still glacierized (cf. Swiss Glacier Inventory SGI2016, Linsbauer et al., 2021), leaving the cirque of Le Dar Dessus largely ice-free. Glacier du Dar disintegrated into two very small remnant ice patches, and Glacier du Sex Rouge (0.26 km² in 2016) retreated above the first rock step and is expected

to have completely disappeared by 2035 (Huss and Fischer, 2016; Fischer and Keiler, 2019).

Due to the rather complex local geology (Badoux and Gabus, 1991), the loose material in the glacier forefield and adjacent areas strongly varies in terms of grain sizes: fine-grained material originating from marly limestones, marly shales and clayey





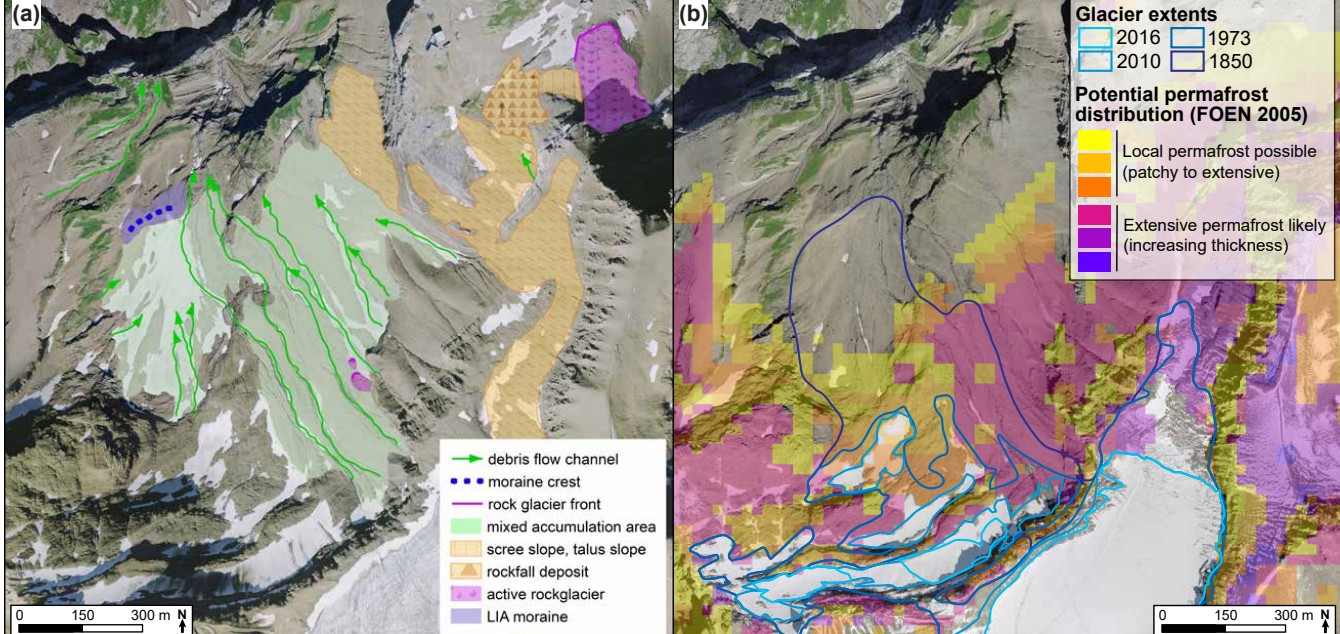

**Figure 2.** (a) Geomorphological characteristics of the debris flow initiation zone (cirque of Le Dar Dessus) around 2005 (modified after Lambiel et al., 2008). Underlain is a 2013 SWISSIMAGE orthophoto from swisstopo. (b) Extract of the 2005 map of potential permafrost distribution in Switzerland (FOEN, 2005). Glacier extents from the Swiss Glacier Inventories SGI1850, 1973, 2010 and 2016 (Müller et al., 1976; Maisch et al., 2000; Paul, 2004; Fischer et al., 2014; Linsbauer et al., 2021). Underlain is a 2020 SWISSIMAGE orthophoto from swisstopo.

limestones, and coarser material (brick size) from siliceous limestones (Helvetic Kieselkalk Formation). Larger blocks originating from the steepest parts of the northwestern flank of Oldehore/Becca d'Audon or from the first rock step (both composed

of Helvetic Kieselkalk) occur as well.

In 2005, about half of the forefield of Glacier du Sex Rouge and adjacent areas was mapped as zones where either local permafrost is possible or extensive permafrost is likely (FOEN, 2005, Fig. 2b). No measurements (e.g., ground surface temperature, bottom temperature of snow cover, geophysical surveys) have so far been carried out to investigate the current permafrost distribution within the cirque of Le Dar Dessus. A protalus rampart below the first rock step and the Entre la Reille rock glacier

at ca. 2450 m a.s.l. (Reynard et al., 1999; Bosson and Lambiel, 2016), just a few hundreds of meters outside of the cirque of Le Dar Dessus (Fig. 2a), however, indicate the probable occurrence of permafrost above ca. 2500 m a.s.l. (Lambiel et al., 2008).

Several gullies (>1 m in depth) cut through the sediment deposits in southeast-northwest direction (Fig. 2a, Fig. A1b) and point to the potential of fluvial erosion during heavy precipitation events. In addition, traces of debris flow initiation of type 1 (in steep and unconsolidated morainic or slope debris) and 2 (at contact zone of a rock wall and steep talus slope) are found

in the cirque of Le Dar Dessus (cf. Rickenmann and Zimmermann, 1993). Despite this clear susceptibility for debris flow



triggering within the cirque, only one major debris flow event associated with flooding in the village of Les Diablerets has been properly documented and occurred in summer 2005 (chapter 2.3).

## 2.3 The summer 2005 debris flow event

On June 24, 2005, a severe thunderstorm event with maximum rainfall intensities of 40 to 100 mm/h hit the area, leading
to peak discharges of 30 to 35 m$^3$/s in the Grande Eau at Les Diablerets (B+C Ingénieurs SA, 2006; Lambiel et al., 2008; Schoeneich and Consuegra, 2008). In the Dar catchment, the heavy rainfall event of an estimated return period between 50 to 100 years triggered a debris flow (cf. official event cadastre of the Canton of Vaud, 2022). The initiation zone of the debris flow was situated at ca. 2450 m a.s.l. in the cirque of Le Dar Dessus, which approximately corresponds to the presumed lower limit of permafrost in 2005 (Fig. 2b and Fig. 3). Humid horizons were visible on the sidewalls of several gullies up to several
weeks after the event, indicating the possible occurrence of ground ice melt. Thus, permafrost degradation might have played a role in the debris flow initiation facilitating the mobilization of recently thawed material (Lambiel et al., 2008). The majority of the total sediment volume transported by the debris flow originated from erosion of (fluvially reworked) glacigenic and gravitational deposits in the cirque (Fig. 3), and was deposited at Creux du Pillon due to the abrupt change in slope of the torrent below the third steep rock step (Cascade du Dar; Fig. 3, Fig. A1e,f). The flood, however, travelled further downvalley
and was reloaded with sediments through bed and bank erosion along a ca. 1.5 km long, more gently inclined section of the Dar (Fig. A1g). Due to further decrease in slope (Fig. 1c, Fig. A1h,i), the material eroded and transported by the flood was deposited again just before the confluence with La Grande Eau (Fig. 3, Schoeneich and Consuegra, 2008). The major damage to infrastructure in Les Diablerets occurring on June 24, 2005 was thus not primarily caused by the debris flow in the Dar catchment, but by subsequent flooding and silt deposits along La Grande Eau downstream from its confluence with Le Dar.
Therefore, processes occurring in the upper catchment of La Grande Eau (Pierredar) adjacent to the Dar catchment also played an important role in the flooding at Les Diablerets (B+C Ingénieurs SA, 2006; Schoeneich and Consuegra, 2008).

Only a few weeks after the first debris flow, on July 29, 2005, another thunderstorm with intense rainfall occurred in the area, causing again a combination of debris flow and flooding in the Dar catchment (cf. official event cadastre of the Canton of Vaud, 2022), but of lesser magnitude compared to June 24. The processes mostly led to erosion and remobilisation of sediment
deposits within the torrent channel, again with a main initiation zone situated in the cirque of Le Dar Dessus between 2000 and 2500 m a.s.l. (Schoeneich and Consuegra, 2008).

No debris flows with magnitudes comparable to the summer 2005 event were recorded in the Dar catchment before or since. However, from the analysis of aerial imagery time series ("A journey through time – aerial images" available through the webpage of the Federal Office of Topography (swisstopo, 2022)) or the comparison of digital terrain models from recent years
(Groos et al., 2022), it can be seen that a number of small(er) events did take place along the uppermost sections of the Dar torrent, both before and after 2005. These were, however, mostly restricted to the cirque of Le Dar Dessus (Fig. 2a, Fig. A1b). At most, they flowed down the second steep rock step and came to a halt shortly thereafter, forming a fan-like landform of intermediate sediment deposits (Fig. 3, Fig. A1c). The lower end of this intermediate fan represents the maximum runout for low magnitude, high frequency events starting in the cirque of Le Dar Dessus (cf. Bracken et al., 2015). The 2005 event can





**Figure 3.** Mapped zones of erosion (red) and deposition (dark blue) of sediments in the course of the summer 2005 debris flow event in the Dar catchment. Most of the sediments transported by the debris flow originated from the cirque of Le Dar Dessus, and were deposited again due to the abrupt change in slope at Creux du Pillon (Fig. 1c), i.e. short after the third steep rock step (Cascade du Dar) and the sharp left turn of the torrent. Fluvial deposits towards the confluence of the Dar with La Grande Eau originated from bed and bank erosion due to high water between Creux du Pillon and Les Diablerets. Flooded areas on both sides of La Grande Eau at Les Diablerets with deposits of fine-grained material are shown in light blue. After Schoeneich and Consuegra (2008). Photographs (24.06. and 10.08.2005): courtesy of B+C Ingénieurs SA (2005). Underlain is a 2020 SWISSIMAGE orthophoto from swisstopo.

be considered as a major and, to our knowledge, to date unique event. It most probably required the exceedance of system thresholds (for both the intensity of the triggering meteorological event and the magnitude of the transported sediment volume) for debris flows to travel beyond the third steep rock step.



## 3   Scenario building for PPDFs

"Scenarios" represent possible future events and sequences of events, for each of which a number of complex interactions
between different meteorological and geomorphological conditions prior to the event, different triggering mechanisms and the
geomorphological processes themselves may occur (e.g., Jakob et al., 2016). Assigned to defined magnitudes and/or return
periods, they serve as an important base for hazard assessment. In Switzerland, scenarios for three (four if the extreme event
is considered as well) return periods are used in the context of hazard mapping (Loat and Petrascheck, 1997). Our approach to
build scenarios for potential PPDFs (Fig. 4) includes the site-specific (i) assessment and definition of possible meteorological
and hydrological triggering events (chapter 3.1), (ii) estimation of sediment volumes (bed load potential and volumes) that
could be mobilised in the course of an event (chapter 3.2), and (iii) definition of total event volumes (water-debris mixtures)
as well as event sequences (number of surges and associated hydrographs, flow behaviour etc.) (chapter 3.3). The final PPDF
scenarios can then be used as input for numerical debris flow simulations (here with RAMMS-DF) to assess flow paths,
flow heights, velocities, transported and deposited sediment volumes and runout distances of potential PPDFs in the studied
catchment (here the Dar, chapter 4).

### 3.1   Definition of possible meteorological and hydrological triggering scenarios

Triggering events for PPDFs can either be precipitation dependent (thunderstorms, long-lasting rainfalls or rain-on-snow (ROS)
events, e.g., Rickenmann and Zimmermann, 1993; Godt and Coe, 2007; Rössler et al., 2014) or precipitation independent (snow
melt, glacier lake outburst flood (GLOF), glacial water pocket outburst flood (WPOF), e.g., Zimmermann, 1990; Bardou and
Delaloye, 2004; Erokhin et al., 2018). For the investigated catchment, potential triggering processes should be assessed and a
range of plausible intensities for each triggering mechanism should be estimated.

For the cirque of Le Dar Dessus, it is known from the 2005 event that heavy thunderstorms can cause debris flows in the area
(chapter 2.3). Based on our knowledge of the (sub)surface characteristics and the snow amounts in spring in the cirque of Le
Dar Dessus, both long-lasting rainfalls and ROS events cannot be excluded as potential triggering processes, either. ROS events
have indeed already been observed and documented in the region, but never in direct connection with subsequent debris flows
(Schoeneich, 1992, 1996). Similarly to long-lasting rainfall events in summer, intense snow melt in spring due to a sudden
warm and dry event (cf. Van Tiel et al., 2021) could potentially lead to water oversaturation of loose material in the area, which
could initiate debris flows through liquefaction of sediments in channel beds.

Specific scenarios for snow melt as an individual debris flow triggering mechanism were not defined here, as it is argued that
the possible range of intensities for snow melt only scenarios is already sufficiently represented by both long-lasting rainfall
scenarios and snow melt dominated ROS scenarios (chapters 3.1.1 to  3.1.3). Due to the rapid shrinkage of Glacier du Sex
Rouge and the formation of a small proglacial lake just above the first rock step (Fischer and Keiler, 2019), also GLOFs were
considered as a potential PPDF triggering mechanism. Assessing the susceptibility and implications of potential future GLOFs
(Fig. B1-B3, Table B1) showed that, in the worst case, a few $10^3$ m$^3$ of rockfall from the northwestern flank of Oldehore/Becca
d'Audon (Fig. 1a,c) could reach the new lake and cause a small outburst of a few $10^2$ to $10^3$ m$^3$ by lake water overtopping



**Figure 4.** Schematic workflow of the developed methodological approach for scenario building and numerical runout modelling of pro-/periglacial debris flows (PPDFs) in catchments with scarce past event data.



the first steep rock step. However, because of the relatively small potential outburst volume and the rather low likelihood of such an event, GLOFs are not considered as relevant PPDF triggering mechanism here. It is expected that such an event would at most result in relocation of loose material from the upper to the lower parts of the cirque of Le Dar Dessus or beyond the second steep rock step, but not directly causing debris flows travelling beyond the third steep rock step. Finally, WPOFs
can be neglected as potential PPDF triggering mechanisms here, as englacial water pockets can only form due to blockage of en- or subglacial macroporosity structures (large crevasses, moulins, en-/subglacial channels) (Haeberli, 1983; Fountain and Walder, 1998). Due to its low dynamics (thin and relatively flat), Glacier du Sex Rouge shows, however, no large crevasses, and meltwater primarily runs off in supraglacial or glacier-marginal channels (Fischer et al., 2016; Fischer, 2018).

### 3.1.1   Thunderstorms and long-lasting rainfalls

To define a plausible range of precipitation scenarios which could potentially trigger future PPDFs, information on quantity, duration and probability of occurrence of extreme rainfalls is needed for the debris flow initiation zone. Here we used data on hourly and daily sums of extreme point rainfall with varying return periods provided by the Hydrological Atlas of Switzerland (HADES) to define different precipitation scenarios for both thunderstorms and long-lasting rainfalls. The data used represents precipitation levels for rare (return period of 2 to 200 years) heavy precipitation events in today's climate. It comes with a
ground resolution of 1 km$^2$ for entire Switzerland, and was derived by extreme value analysis of precipitation data from a large number of MeteoSwiss weather stations and subsequent spatial interpolation (Frei and Fukutome, 2022). Hourly (respectively daily) sums of extreme point rainfall for the area of the cirque of Le Dar Dessus with a return period of 2 years were assigned to the "small", with a return period of 30 years to the "medium", with a return period of 100 years to the "large", and with a return period of 200 years to the "very large" thunderstorm (respectively long-lasting rainfall) scenario.

To approximately represent the course of a typical thunderstorm, the values of hourly precipitation sums (mm) were linearly interpolated to five minutes intervals, so that maximum rainfall intensities are reached after 30 minutes and linearly decrease again after that. Total hourly precipitation sums vary according to the scenario size between 23 and 72 mm, with maximum intensities between 45 and 139 mm/h (Table 1). The thunderstorm of June 24, 2005 with reported precipitation sums of >100 mm within two hours and estimated intensities of 40 to 100 mm/h (chapter 2.3, B+C Ingénieurs SA, 2006) would be assigned
to a medium to large event here. Following Zimmermann et al. (1997a), the critical mean rainfall intensity necessary to trigger debris flows by thunderstorms is 43 * 1$^{-0.89}$ = 43 mm/h. This threshold is exceeded for all thunderstorm scenarios except for the small scenario, for which precipitation intensities >43 mm/h prevail only during 5 minutes (Table 1).

For the long-lasting rainfall scenarios, the daily values of extreme point rainfall were linearly interpolated to hourly intervals, so that constant intensities prevail during 24 hours. As for these scenarios the duration of the precipitation events matters more than the intensity, assuming constant hourly rainfall intensities was judged to be justified, even though it represents a
simplification of the reality. Precipitation sums range from 93 mm/d for the small scenario to 214 mm/d for the very large scenario, with corresponding (mean) intensities ranging from 3.9 (small scenario) to 8.9 mm/h (very large scenario) (Table 1). According to Zimmermann et al. (1997a), the critical mean rainfall intensity necessary to trigger debris flows by long-lasting



**Table 1.** Thunderstorm and long-lasting rainfall scenarios for the cirque of Le Dar Dessus. Five minutes precipitation intensities and hourly precipitation sums are listed for the thunderstorm scenarios. Maximum intensities after 30 minutes are highlighted in bold. Note that hourly precipitation sums (mm) equal mean hourly precipitation intensities (mm/h). Hourly precipitation intensities and daily precipitation sums are listed for the long-lasting rainfall scenarios. As a simplifying assumption, precipitation intensities are held constant over one day.

| Thunderstorm scenario | Precipitation intensity (mm/h) | | | | | | | | | | | | | Hourly precipitation sum (mm) |
|---|---|---|---|---|---|---|---|---|---|---|---|---|---|---|
| Time (minutes) | 0 | 5 | 10 | 15 | 20 | 25 | 30 | 35 | 40 | 45 | 50 | 55 | 60 | |
| small | 4.5 | 11.9 | 19.3 | 26.7 | 34.1 | 41.5 | **44.5** | 41.5 | 34.1 | 26.7 | 19.3 | 11.9 | 4.5 | 23 |
| medium | 9.5 | 25.3 | 41.1 | 56.9 | 72.7 | 88.5 | **94.8** | 88.5 | 72.7 | 56.9 | 41.1 | 25.3 | 9.5 | 49 |
| large | 12.2 | 32.5 | 52.8 | 73.2 | 93.5 | 113.8 | **121.9** | 113.8 | 93.5 | 73.2 | 52.8 | 32.5 | 12.2 | 63 |
| very large | 13.9 | 37.2 | 60.4 | 83.6 | 106.8 | 130.1 | **139.4** | 130.1 | 106.8 | 83.6 | 60.4 | 37.2 | 13.9 | 72 |
| Long-lasting rainfall scenario | Precipitation intensity (mm/h) | | | | | | | | | | | | | Daily precipitation sum (mm) |
| small | | | | | | | 3.9 | | | | | | | 93 |
| medium | | | | | | | 6.8 | | | | | | | 163 |
| large | | | | | | | 8.1 | | | | | | | 195 |
| very large | | | | | | | 8.9 | | | | | | | 214 |

rainfalls is $43 * 24^{-0.89} = 2.54$ mm/h (corresponding to a daily sum of 61 mm). This threshold is exceeded for all long-lasting

rainfall scenarios here (Table 1).

### 3.1.2 Rain-on-snow events

ROS events are relevant for the triggering of debris flows in high mountain areas, as the combination of snow melt, saturated underground and rainfall can lead to extraordinarily high runoff in torrent catchments (e.g., Rössler et al., 2014). Predicting the influence of the snow cover on runoff generation during heavy precipitation events is, however, very challenging, as processes

involved in ROS events are highly complex and may vary considerably over small spatiotemporal scales (Würzer, 2018).

Based on our knowledge of the Dar catchment from over ten years of seasonal mass balance monitoring on Glacier du Sex Rouge (GLAMOS, 2021), ROS events that could potentially trigger debris flows in the cirque of Le Dar Dessus are expected to happen in late spring or early summer (May-July). As snow cover can intermittently store significant amounts of rain and therefore has a dampening effect on intensity and runoff generation from short and intense rainfalls (e.g., Seibert



**Table 2.** Derivation of snow melt intensities and snow cover runoff for both ROS events dominated by rain and snow melt.

| Parameter | Proportion of rain dominates in the snow cover runoff | Proportion of snow melt dominates in the snow cover runoff |
| --- | --- | --- |
| Rainfall intensity (mm/h) | Same as in Table 1 for long-lasting rainfalls | Same as in Table 1 for long-lasting rainfalls |
| Snow melt intensity (mm/h) | Rainfall intensity * 3/7 | Rainfall intensity * 7/3 |
| Snow cover runoff (mm/h) | 70% rainfall intensity + 30% snow melt intensity | 30% rainfall intensity + 70% snow melt intensity |

et al., 2014; Würzer and Jonas, 2017), thunderstorms are considered not being able to cause ROS events with subsequent triggering of debris flows in the cirque of Le Dar Dessus. Therefore, the same precipitation scenarios as derived for long-lasting rainfalls (chapter 3.1.1, Table 1) were used to estimate snow cover runoff for different ROS scenarios. Moreover, a simple but straightforward approach was chosen to best possibly represent the range of plausible snow cover runoff scenarios for future potential ROS events. Two different classes of ROS scenarios were defined, one for which the proportion of rain

dominates in the runoff and one for which the proportion of snow melt dominates. Following Würzer and Jonas (2017), snow melt intensities were calculated as quotients of the long-lasting rainfall intensities (Table 2).

No measured data on the spatial distribution of snow depths or snowpack characteristics are available for the cirque of Le Dar Dessus. In close proximity to the debris flow initiation zone, however, continuous seasonal in-situ mass balance measurements are available for Glacier du Sex Rouge since 2012 (Fischer, 2018; GLAMOS, 2021). Based on time series of measured snow

depths and density profiles on the glacier, average snow water equivalent (SWE) in the (upper parts of the) cirque of Le Dar Dessus at the time of maximum end-of-winter snow cover was estimated to 2000 mm. This value was adopted here as an upper limit average for late spring to early summer in order to calculate snow cover runoff intensities and clear water discharges for the different ROS scenarios which could potentially trigger debris flows. For simplification and due to the lack of spatially distributed data, the same average SWE was assigned to the entire cirque of Le Dar Dessus.

Following Würzer and Jonas (2017), it was further assumed here for all ROS scenarios that the snowpack absorbs water from rainfall during the first six hours of the precipitation event, implying a negative snow cover excess runoff as well as a slight rise in SWE through long-lasting rainfalls during that time (Table 3). Six hours after the onset of the long-lasting rainfalls, snow cover excess runoff turns into positive values again, implying a reduction in SWE. Snow melt water only then starts to contribute to runoff (Fig. 5 for the very large ROS scenarios).

For rain dominated ROS scenarios, total snow cover runoff (over 24 h) varies between 100 and 229 mm according to the scenario size (Table 3), thus values comparable to the daily precipitation sums of the different long-lasting rainfall scenarios (Table 1). Snow melt dominated ROS scenarios produce significantly higher total snow cover runoff of between 234 and 534 mm (over 24 h) depending on the scenario size (Table 3).





**Table 3.** Rainfall intensities and snow cover runoff for rain dominated ROS scenarios (snow cover runoff = 70% rainfall + 30% snow melt) and snow melt dominated ROS scenarios (snow cover runoff = 30% rainfall + 70% snow melt).

| Scenario | Rainfall intensity for 24 h (mm/h) | ROS rain dominated | | | | ROS snow melt dominated | | | |
|---|---|---|---|---|---|---|---|---|---|
| | | Snow cover excess runoff | | Snow cover runoff | | Snow cover excess runoff | | Snow cover runoff | |
| | | first 6 h (mm/h) | remaining 18 h (mm/h) | remaining 18 h (mm/h) | total (mm) | first 6 h (mm/h) | remaining 18 h (mm/h) | remaining 18 h (mm/h) | total (mm) |
| small | 3.9 | -3.9 | 1.7 | 5.6 | 100 | -3.9 | 9.1 | 13.0 | 234 |
| medium | 6.8 | -6.8 | 2.9 | 9.7 | 175 | -6.8 | 15.9 | 22.7 | 408 |
| large | 8.1 | -8.1 | 3.5 | 11.6 | 208 | -8.1 | 18.9 | 27.0 | 486 |
| very large | 8.9 | -8.9 | 3.8 | 12.7 | 229 | -8.9 | 20.8 | 29.7 | 534 |

### 3.1.3 Calculation of clear water discharge per triggering scenario

If no measured runoff data are available for a catchment, clear water discharge associated with the previously defined precipitation dependent triggering events can be estimated based on empirical relations or precipitation-runoff models (Kölla, 1987; Blöschl, 2006). Clear water discharge in torrents actually never occurs in reality during floods or debris flows, as significant amounts of loose material are typically transported in the course of such events. However, estimating clear water discharges for each triggering scenario is helpful to assign the latter to different debris flow scenarios and assess their probability of occurrence (chapter 3.3). Moreover, it can be used to define the downstream implications of flooding and bed load transport in case no debris flow is triggered.

For the present study, version 2.0 of ZEMOKOST (ZEller MOdifiziert von KOhl und STepanek, developed at the Department for Natural Hazards at the Austrian Research Centre for Forests (BFW)), a relatively simple, freely available, Excel-based precipitation-runoff model for small and unmeasured torrent catchments (Kohl and Stepanek, 2005; Kohl, 2010), was used to calculate clear water discharges for all thunderstorm, long-lasting rainfall and ROS scenarios. ZEMOKOST is based on a modified runtime procedure. Depending on the transit time of water at the surface (calculated with the Izzard formula) and in the torrent channel (calculated using the Manning-Strickler formula), runoff hydrographs caused by defined precipitation events can be calculated for sub-catchments or specific locations in the channel. The amount of rain that contributes to runoff results from the total precipitation sum multiplied by a runoff coefficient. Runoff coefficients define the relative amount of precipitation that directly runs off as surface flow. They vary according to different surface characteristics and soil properties, and have to be assessed, together with other parameters such as catchment size, channel length and slope, vegetation cover etc., in order to run the model.



**Figure 5.** Evolution of snow water equivalent, hourly rainfall and snow melt intensities as well as snow cover runoff for (a) the very large rain dominated ROS scenario and (c) the very large snow melt dominated ROS scenario. The evolution of snow cover excess runoff is shown in (b) for the very large rain dominated ROS scenario and in (d) for the very large snow melt dominated ROS scenario.

Hydrographs for clear water discharge were calculated for a point in the torrent channel in the middle of the second steep rock step (Fig. 1a,c, Fig. C1), where the individual channels developing within the cirque of Le Dar Dessus come together.

The necessary input parameters for ZEMOKOST describing catchment characteristics (area, slope, hillslope length, baseflow, sediment grain size, runoff and roughness coefficients for different substratum) were derived based on assessments in the field and using most recent aerial orthophotos, DEMs and topographic maps available from swisstopo (Fig. C1, Table C1). They were held constant for all scenarios, except for the baseflow and the runoff coefficients (Table C1).

Based on the defined intensities for thunderstorm and long-lasting rainfall scenarios as well as snow cover runoff intensities

for the ROS scenarios (chapters 3.1.1 and 3.1.2), clear water discharges and hydrographs resulting from the calculations with ZEMOKOST were derived (Table 4 and Fig. 6). The highest peak discharges are to be expected for thunderstorm scenarios





**Table 4.** Scenario dependent peak clear water discharges and total clear water runoff calculated with ZEMOKOST for the Dar torrent below the cirque of Le Dar Dessus.

| | Peak clear water discharge (m³/s) | | | |
|---|---|---|---|---|
| Scenarios | small | medium | large | very large |
| Thunderstorm | 3.6 | 9.8 | 15.1 | 18.9 |
| Long-lasting rainfall | 1.9 | 2.9 | 3.4 | 3.6 |
| ROS rain dominated | 3.9 | 6.5 | 7.7 | 8.3 |
| ROS snow melt dominated | 8.5 | 14.1 | 16.6 | 18.2 |
| | Total clear water runoff (m³) | | | |
| Scenarios | small | medium | large | very large |
| Thunderstorm | 15'700 | 51'900 | 72'200 | 85'100 |
| Long-lasting rainfall | 111'700 | 197'500 | 236'900 | 261'500 |
| ROS rain dominated | 196'700 | 364'000 | 435'900 | 477'300 |
| ROS snow melt dominated | 488'600 | 853'200 | 1'014'800 | 1'116'300 |

and snow melt dominated ROS scenarios. Total runoff is – due to the different duration of the precipitation events – smallest for thunderstorm scenarios, and significantly increases gradually from long-lasting rainfall scenarios to rain dominated and finally snow melt dominated ROS scenarios (Table 4). For thunderstorm scenarios, peak discharge is to be expected around 75 minutes after the onset of the precipitation event. Hydrographs rise rapidly and steeply and fall again just as quickly after reaching peak discharge. For long-lasting rainfall scenarios (note the simplifying assumption of constant rainfall intensities), peak discharges are almost already reached after about two hours. Clear water discharges only slightly increase after that, until it stops to rain and values decrease again at the end of the precipitation event. For ROS scenarios, hydrographs of clear water discharge are somewhat similar to those of the long-lasting rainfall scenarios (as underlying precipitation scenarios are the same). They show, however, significantly higher peak discharges, slightly less steep rises and drops of runoff intensities, and shorter runoff duration. Due to the intermittent precipitation storage capacity of the snowpack, runoff intensities only begin to rise about seven hours after the onset of the precipitation event (Fig. 6).

### 3.2 Estimation of bed load potential and volumes and definition of bed load scenarios

An important step in scenario building for PPDFs (Fig. 4) relates to the estimation of the bed load potential and volumes for the debris flow initiation zone and the subsequent definition of bed load scenarios (here for small, medium, large and very large events). The bed load potential of a debris flow is equal to the volume of loose material in a catchment that is potentially mobilized during the course of an event. The bed load volume, on the other hand, is the volume of debris that actually passes any specific location in the torrent during a debris flow event (Hunziker et al., unpublished). For PPDFs starting in the cirque







**Figure 6.** Scenario dependent hydrographs for clear water discharge (blue lines) calculated with ZEMOKOST for the Dar torrent below the cirque of Le Dar Dessus. (a-d) Thunderstorm scenarios, (e-h) long-lasting rainfall scenarios, (i-l) rain dominated ROS scenarios, (m-p) snow melt dominated ROS scenarios. Blue bars/areas show the intensities of the underlying precipitation scenarios.

of Le Dar Dessus, a set of plausible bed load scenarios was defined by application of empirical methods and assessments from
field surveys.

According to calculations with empirical formulas most frequently used in practice (Rickenmann, 1999), bed load potentials for PPDFs starting in the cirque of Le Dar Dessus range between 41'000 and 161'000 m$^3$ (average: 110'000 m$^3$, median: 124'000 m$^3$, Table 5). A more detailed empirical approach proposed by Spreafico et al. (1996) for the estimation of bed load volumes in small and steep torrent catchments like the Dar was applied, too (Fig. D1, Tables D1-D2). In contrast to empirical




formulas, this method relies on catchment-specific assessments and knowledge acquired during field surveys, and allows the determination of bed load volumes for a range of events of different magnitude. Following Spreafico et al. (1996), resulting bed load volumes of 35'000, 55'000 and 115'000 m³ were derived for small, medium and large PPDFs starting in the cirque of Le Dar Dessus (Table 5). To complement and for plausibility check of the aforementioned estimations of bed load potentials and bed load volumes, SEDEX© (SEDiments and EXperts, Frick et al., 2011), a practice-oriented assessment tool for field-

based estimations of event-specific bed load volumes in small torrent catchments, was used (Fig. D2). Applying SEDEX©, the resulting bed load volume potentially released in the course of a large debris flow event starting in the cirque of Le Dar Dessus was 125'000 m³ (Table 5), thus similar to the value for a large event derived following Spreafico et al. (1996).

For the final bed load scenarios (Table 6), 35'000 m³ (i.e. the lower limit of estimated bed load volumes, Table 5) was assigned to a small event, and 125'000 m³ (from application of SEDEX©, Table 6) to a large event. The average of these

volumes, i.e. 80'000 m³, was assigned to a medium-sized event. It is argued that this value better covers the range of possible bed load scenarios than the medium bed load scenario of 55'000 m³ derived using the approach by Spreafico et al. (1996). For the very large event, 300'000 m³ was defined for the final bed load scenarios (Table 6). Even though the total volume of loose material currently stored in the cirque of Le Dar Dessus is not known, field observations showed that this number is likely in the order of a few 10⁶ m³. An immediate release of 300'000 m³ in a single event would – in our opinion – correspond to

an "extreme" and rather unlikely event (cf. Loat and Petrascheck, 1997). However, it can be seen as the upper limit of "the potentially possible", which is required for scenario building and hazard assessment.

### 3.3   Definition of total event volumes and debris flow scenarios

As a final step in scenario building for PPDFs (Fig. 4), total event volumes (i.e. volumes of water-debris-mixture released during an event) have to be defined for the different bed load scenarios (chapter 3.2). Additionally, plausible debris flow scenarios have

to be formulated, including the assignment of meteorological and hydrological triggering scenarios (chapter 3.1) to the different event volumes.

The volumetric solids concentration ($c_v$) of alpine debris flows typically varies between 30 and 60% (Iverson et al., 1997). For very steep torrent catchments, however, $c_v$ tends towards the solids concentration of resting material c* (solids concentration limit of debris flows), which in such cases is equal to 0.81 (Weber and Rickenmann, 2000). For torrents with slopes >36%

(respectively >40%) such as the uppermost parts of the Dar, $c_v$ equals to 0.9 * c* (i.e. $c_v \approx 0.7$, cf. Takahashi, 1991; Tognacca, 1999). Thus, the scenario-based total event volumes (hereafter abbreviated with "DF") of PPDFs starting in the cirque of Le Dar Dessus were assumed to consist of maximum 70% debris and minimum 30% water. For simplicity, we divided the total event volumes into two thirds of debris and one third of water. Therefore, total event volumes were calculated by multiplying bed load volumes for different scenarios (chapter 3.2) by a factor of 1.5. Resulting potential event volumes (rounded to 5'000

m³) range between 55'000 (DF$_{small}$) and 450'000 m³ (DF$_{very large}$) according to the scenario size (Table 6).

Following Mizuyama et al. (1992), classes of similar clear water discharge (peak values and duration, Fig. 6) derived from the previously defined meteorological triggering scenarios were used to assign plausible triggering events to the different total event volumes (magnitudes DF$_{small}$ to DF$_{very large}$ in Table 6) of PPDF scenarios as well as to assess their probability of



**Table 5.** Estimated bed load potentials (derived from empirical formulas) and bed load volumes (BL, based on field surveys and estimations for bed load volumes in steep and small torrent catchments) for PPDFs starting in the cirque of Le Dar Dessus. Resulting numbers from the application of the empirical formulas are rounded to the nearest thousand, from the application of bed load volume estimation approaches to the next higher five thousand.

| Method/author | Empirical formula used | Resulting bed load potential ($m^3$) | Remarks |
|---|---|---|---|
| Kronfellner-Kraus (1984) | $V = K * E * J_c$ | 161'000 | V: bed load potential ($m^3$) |
| | $(K = 1750/E^{0.018 * E} = 1701.0)$ | | K: coefficient (-) |
| Zeller (1985) | $V = K * E^{0.78}$ | 41'000 | E: catchment area (2.1 $km^2$) |
| | $(K = 23'000)$ | | $J_c$: slope of main channel (45%) |
| D'Agostino (1996) nr 1 | $V = 45 * E^{0.9} * J_c^{1.5} * IG$ | 132'000 | $J_k$: slope of accumulation area (8%) |
| | | | L: length of main channel from initiation zone to accumulation area |
| D'Agostino (1996) nr 2 | $V = 39 * E * J_c^{1.5} * IG * IT^{-0.3}$ | 124'000 | (3300 m) |
| | | | IG: Geology factor (5 for moraine material and gravel) |
| Rickenmann (1995) | $V = (6.4 * J_k - 23) * L$ | 93'000 | IT: 1 for debris flows |
| **Method/author** | | **Resulting bed load volume ($m^3$)** | **Remarks** |
| Estimating bed load in steep torrents | | $BL_{small}$: 35'000 | $BL_{small}$: 0.583 * $BL_{medium}$ |
| Spreafico et al. (1996), in: Hunziker et al. (2014) | | $BL_{medium}$: 55'000 | $BL_{medium}$: 2 x 27'000 $m^3$ |
| | | $BL_{large}$: 115'000 | $BL_{large}$: 2.083 * $BL_{medium}$ |
| | | | see Fig. D1, Tables D1-D2 |
| SEDEX© (Frick et al., 2011) | | $BL_{large}$: 125'000 | Eastern segment of the cirque of Le Dar Dessus: 55'000 $m^3$ |
| | | | Western segment of the cirque of Le Dar Dessus: 70'000 $m^3$ |
| | | | see Fig. D2 |

occurrence (Table 7). As the critical mean rainfall intensities required to trigger debris flows (Zimmermann et al., 1997a) are
exceeded for all meteorological triggering scenarios except for the small thunderstorm scenario (chapter 3.1, Table 1, Fig. 6), it
is argued that the smallest debris flow scenario ($DF_{small}$) with a total event volume of 55'000 $m^3$ could potentially be triggered
by medium and larger thunderstorm scenarios or by all long-lasting rainfall and ROS scenarios (Table 7). The probability of



**Table 6.** Final bed load scenarios and corresponding total event volumes (DF, rounded to 5'000 m$^3$) for PPDFs starting in the cirque of Le Dar Dessus.

| Bed load scenario | Volume (m$^3$) | Multiplication factor for water content | Event volume (m$^3$) |
|---|---|---|---|
| $BL_{small}$ | 35'000 | 1.5 | $DF_{small}$: 55'000 |
| $BL_{medium}$ | 80'000 | 1.5 | $DF_{medium}$: 120'000 |
| $BL_{large}$ | 125'000 | 1.5 | $DF_{large}$: 190'000 |
| $BL_{very\ large}$ | 300'000 | 1.5 | $DF_{very\ large}$: 450'000 |

occurrence for such an event is judged highest among all four final PPDF scenarios. The small debris flow scenario is most comparable to the 2005 event, to which a return period of 50 to 100 years was assigned (chapter 2.3). However, because both

the susceptibility and probability of occurrence of debris flows starting in the cirque of Le Dar Dessus are expected to increase in the future (Beniston et al., 2018), it is probably more realistic to assign a return period of 30 to 50 years to the small debris flow scenario ("it could happen once in one or two generations").

Compared to the small scenario, triggering the medium debris flow scenario ($DF_{medium}$) with a total event volume of 120'000 m$^3$ would require more intense thunderstorm or long-lasting rainfall scenarios to happen (Table 7). Being comparable or higher

than the very large long-lasting rainfall scenario (Table 4, Fig. 6), the runoff (intensities and volumes) generated from the smallest ROS scenarios (and thus from all ROS scenarios) would probably be sufficient to trigger the medium debris flow scenario (Table 7). The latter is assigned to a return period of about 100 years ("it could happen once in a lifetime").

The large debris flow scenario ($DF_{large}$) with a total event volume of 190'000 m$^3$ would be associated to an event with "historical dimensions" (i.e. with an estimated return period of several hundred years). It is assumed that intensities of debris

flow triggering meteorological and hydrological events would have to significantly exceed those potentially able to trigger a medium debris flow scenario. Therefore, it is argued that only the very large thunderstorm scenario as well as all ROS scenarios (except the small rain dominated one) could potentially trigger the large debris flow scenario (Table 7). As runoff intensities of the very large long-lasting rainfall scenario are comparable to the small rain dominated ROS scenario (Table 4, Fig. 6), none of the long-lasting rainfall scenarios for the cirque of Le Dar Dessus is thought to be able to potentially trigger a debris flow

with a total event volume of 190'000 m$^3$.

The very large debris flow scenario ($DF_{very\ large}$) with a total event volume of 450'000 m$^3$ is judged as being (highly) unlikely to happen. It can be seen as the "extreme scenario", at the upper limit of the potentially possible. Intensities of debris flow triggering meteorological and hydrological events being able to potentially trigger such an extreme scenario would have to be at the upper limit of the potentially possible as well (Table 4, Fig. 6). Therefore, it is argued that only (medium or larger) snow

melt dominated ROS events could potentially trigger the very large debris flow scenario (Table 7).





**Table 7.** Assignment of precipitation dependent meteorological (Table 1) and hydrological (Fig. 6) scenarios to debris flow scenarios (total event volumes, Table 6) for the cirque of Le Dar Dessus. Crosses stand for scenario combinations assessed as potentially possible. In addition, a qualitative probability of occurrence for each debris flow scenario is given.

| Total event volume | | Debris flow triggering scenarios | | | | | | | | | | | | | | | | Probability of occurrence |
|---|---|---|---|---|---|---|---|---|---|---|---|---|---|---|---|---|---|---|
| | | Thunderstorm | | | | Long-lasting rainfall | | | | Rain dominated ROS | | | | Snow melt dominated ROS | | | | |
| Scenario | Volume (m³) | small | medium | large | very large | small | medium | large | very large | small | medium | large | very large | small | medium | large | very large | |
| DF$_{small}$ | 55'000 | - | X | X | X | X | X | X | X | X | X | X | X | X | X | X | X | "once in one or two generations" (i.e. return period of 30 to 50 years) |
| DF$_{medium}$ | 120'000 | - | - | X | X | - | X | X | X | X | X | X | X | X | X | X | X | "once in a lifetime" (i.e. return period of 100 years) |
| DF$_{large}$ | 190'000 | - | - | - | X | - | - | - | - | - | X | X | X | X | X | X | X | "historical event" (i.e. return period of several hundred years) |
| DF$_{very\ large}$ | 450'000 | - | - | - | - | - | - | - | - | - | - | - | - | - | X | X | X | (highly) unlikely, "extreme scenario" |

After defining total event volumes and assigning them to meteorological and hydrological triggering events for each PPDF scenario, the latter can be used as input for debris flow runout modelling, in our case using RAMMS-DF. For that purpose, event sequences (sub-scenarios) including a number of surges with associated volumes, hydrographs and flow behaviour need to be defined for each scenario. Here, plausible sub-scenarios rely on (i) the reconstruction of the summer 2005 debris flow event (cf. chapter 4.1) and (ii) the assumption that future debris flows would behave similarly to this only recorded major debris flow event. For better traceability, the final debris flow sub-scenarios are described at the beginning of chapter 4.2 and listed in Table 12 after the calibration of RAMMS-DF to the Dar.

## 4  Modelling PPDF scenarios for the Dar catchment using RAMMS-DF

The RAMMS::Debrisflow module (RAMMS-DF) from the software package RAMMS (Rapid Mass Movement Simulation) was used in this study to simulate the runout of the debris flow scenarios defined for the Dar catchment. RAMMS-DF is a two-dimensional, physically based numerical model developed by experts at the Swiss Federal Institute for Forest, Snow and Landscape Research (WSL) to calculate three-dimensional and slope-parallel velocities and flow heights of debris flows from initiation to runout (Bartelt et al., 2017). It is based on the implementation of the two-parameter Voellmy relation to describe the





frictional behaviour of debris flows. This relation separates the total basal friction into a velocity independent Coulomb term
for dry friction ($\mu$) and a velocity dependent term for turbulent friction ($\xi$) (Hussin et al., 2012). In order to run RAMMS-DF, the two main friction parameters $\mu$ and $\xi$ have to be defined by the user. They should be calibrated for individual torrents or catchments using – if available – information and data about well-documented past events (Deubelbeiss and Graf, 2013). For the Dar, $\mu$ and $\xi$ were calibrated using data from the 2005 event (Fig. 4; chapter 4.1). Subsequently, future potential PPDFs could be simulated based on the previously defined scenarios (chapter 4.2).

### 4.1 Calibration of RAMMS-DF to the 2005 debris flow event

Before modelling PPDF scenarios, mapped extents of debris flow deposits from the summer 2005 event (Fig. 3, Schoeneich and Consuegra, 2008) were used to calibrate RAMMS-DF to the Dar catchment. First, the model was run based on estimations of the total 2005 debris flow volume and initial (roughly estimated) values for the friction parameters. The resulting modelled debris flow deposition extent was then tested against the known extent of the 2005 event. Both the total event volume and the
two main friction parameters were adjusted in an iterative way until the best representation of the 2005 debris flow extent was reached (Fig. 4).

#### 4.1.1 Extent of debris flow deposits and maximum flow heights of the 2005 event

Schoeneich and Consuegra (2008) analysed the summer 2005 thunderstorms and subsequent flood and debris flow event in both the Dar and Grande Eau catchments and mapped areas of (channel) erosion and debris (flow) deposits (Fig. 3). Before
being used for the calibration of RAMMS-DF, the digitized extents of the 2005 debris flow deposits at Creux du Pillon were slightly adjusted after comparison with extensive photo documentation from August 10, 2005 (courtesy of B+C Ingénieurs SA, 2005) and high-resolution (25 cm) SWISSIMAGE Level 2 orthophotos from 2010 available from swisstopo. The adjusted digitized extent of the 2005 debris flow deposits is shown in Fig. 7a.

     The calibration of RAMMS-DF to the 2005 debris flow event would also have benefited from information about flow heights
and flow velocities (Fig. 4), since these parameters can be easily analyzed for different model runs. Flow velocities were, however, unfortunately not recorded, nor could they be reconstructed from existing data or documentation of the summer 2005 event. Based on photo documentation from August 10, 2005 (courtesy of B+C Ingénieurs SA, 2005), maximum debris flow height in the middle of the accumulation zone (channel at CSP3, Fig. 7a,b) was estimated to 1.6 m, and to 2 m at maximum runout of the 2005 debris flow (hiking trail bridge at CSP4, Fig. 7a,c,d). However, due to the lower quality and amount of
information about flow heights compared to the digitized extents of the 2005 debris flow deposits, it was decided to carry out the calibration of RAMMS-DF to the 2005 debris flow event only based on the known extent of the debris flow deposits.

#### 4.1.2 Input DEM

To calibrate RAMMS-DF to the 2005 event, a DEM reflecting the topography of both the bed of the Dar torrent and adjacent areas before June 2005 is needed. Therefore, a high-resolution (1 m) DEM created from consecutive airborne laserscanning







**Figure 7.** (a) Digitized extents of the summer 2005 debris flow deposits at Creux du Pillon, slightly adjusted from Schoeneich and Consuegra (2008). Flooded areas without deposits are not shown. Underlain is a 2013 SWISSIMAGE orthophoto from swisstopo. (b-d) The channel of the Dar at two cross sections (CSP3 and CSP4, see (a)) on August 10, 2005, and estimated flow heights of the 2005 debris flow (photographs: courtesy of B+C Ingénieurs SA, 2005). Blue arrows indicate the flow direction of the Dar.

(ALS) surveys of 2001–2002 (for areas below 2000 m a.s.l.) and 2005 (for areas >2000 m a.s.l.) was used here (Table 8). The vertical accuracy of the ALS DEM ranges between ±0.5 m for open terrain and ±1.5 m for areas of dense vegetation cover (Office de l'information sur le Territoire, 2002).



**Table 8.** Different DEMs used in this study for debris flow modelling with RAMMS-DF.

| DEM source | Acquisition type | Acquisition date | Resolution | Vertical accuracy | Purpose of use |
|---|---|---|---|---|---|
| Office de l'information sur le territoire, Canton of Vaud | Airborne laserscanning | Between January 17 and June 4, 2001 and between May 14 and October 21, 2002 (areas <2000 m a.s.l.); October 2005 (areas >2000 m a.s.l.) | 1 m | ±0.5 m (open terrain); ±1.5 m (areas of dense vegetation cover) | Calibration of RAMMS-DF to the summer 2005 debris flow event |
| swissALTI$^{3D}$ release 2018, Federal Office of Topography swisstopo | Airborne laserscanning (areas <2000 m a.s.l.); stereocorrelation of high-resolution aerial orthophotos (areas >2000 m a.s.l.) | June to August 2016 | 0.5 m | ±0.5 m (areas <2000 m a.s.l.); ±1 to 3 m (areas >2000 m a.s.l.) | Modelling debris flow scenarios for the Dar catchment with RAMMS-DF |

### 4.1.3 Definition of initial values for the friction parameters

Prior to iterative model calibration, initial values had to be chosen for both friction parameters. The initial value for $\mu$ (dry
friction) can be derived from the tangent of the slope angle ($\tan(\alpha)$) of the debris flow deposits in the accumulation area (Bartelt
et al., 2017). Based on the mean channel slope of 4.65° (8.13%) in the accumulation area at Creux du Pillon (Fig. 1a,c, Fig. 3),
$\mu$ was set to 0.08. The initial value of $\xi$ (turbulent friction) depends on the flow behaviour. For granular debris flows, $\xi$ values
of 100–200 m/s$^2$ are typical, whereas for mud flows, $\xi$ ranges between 200 and 1000 m/s$^2$ (Bartelt et al., 2017). Due to the
grain size distribution of the unconsolidated sediments in the cirque of Le Dar Dessus (predominantly coarse material but with
a significant proportion of fine-grained sediments, chapter 2.2), rather granular debris flows (and hence $\xi$ values between 100
and 200 m/s$^2$) were first expected for the Dar. Because of the steep terrain in the upper part of the torrent (chapter 2.1), however,
relatively high flow velocities have to be assumed, implying in turn higher $\xi$ values. Therefore, calibration of RAMMS-DF was
started with an initial $\xi$ value of 200 m/s$^2$.

### 4.1.4 Input hydrograph

As recommended for large or channelized debris flows (Deubelbeiss and Graf, 2013; Bartelt et al., 2017), simulations with
RAMMS-DF for both model calibration and modelling of PPDF scenarios were started using simplified 3-point input hydro-
graphs. Input hydrographs are derived from empirical relations between the total released volume and the maximum discharge



of the debris flow. Table 9 shows how the different parameters of the input hydrograph used to calibrate RAMMS-DF were derived. The location of the input hydrograph, i.e. the starting point for debris flow modelling with RAMMS-DF, was set at 1795 m a.s.l. in the channel of the Dar (light blue triangle in Fig. 1a), just below the second steep rock step and thus more than 1000 m (horizontal distance) and 650 m (vertical drop) further downvalley compared to the estimated initiation zone of the 2005 event (Lambiel et al., 2008; Schoeneich and Consuegra, 2008). This choice was driven by the fact that (i) a DEM with satisfactory quality and resolution acquired before the summer 2005 event is only available for areas below 2000 m a.s.l. (Table 8), and (ii) the chosen starting point is where all headwaters, channels and gullies of the upper Dar catchment flow together and concentrate in one single channel (Fig. 1a).

Unfortunately, the total debris flow volume (water-debris mixture) released in the cirque of Le Dar Dessus during the summer 2005 event was not reported. Likewise, no estimates of the deposited sediment volume at Creux du Pillon were made shortly after the event. Schoeneich and Consuegra (2008) estimated the total volume of loose material mobilized in the course of the thunderstorms on June 24 and July 29, 2005 in both the Dar and Grande Eau catchments to more than 100'000 m$^3$. Calibration of RAMMS-DF to the 2005 event was therefore started with the initial values for $\mu$ and $\xi$ (chapter 4.1.3) and an input hydrograph of $V$ = 125'000 m$^3$, representing an uppermost limit of possible debris flow volume for the 2005 event. However, with this released volume, the modelled extent of the debris flow deposits turned out to be far too large compared to the 2005 deposits (Fig. 7a). Therefore, the total released volume was adjusted iteratively through several model runs with successively decreasing $V$ but constant initial values for $\mu$ and $\xi$, and validated against the digitized extents of the 2005 debris flow deposits. The result of this iterative approximation of $V$ showed that the area of the 2005 debris flow deposits is best represented with a total released debris flow volume of about 30'000 m$^3$. This is less than reported deposited volume estimates of 46'500 m$^3$ at Creux du Pillon in summer 2005 (as reconstructed by practitioners in the field in 2012, B+C Ingénieurs SA, 2013).

### 4.1.5   Additional parameters

In addition to the use of a suitable DEM, the definition of an input hydrograph as well as friction parameters $\mu$ and $\xi$, the input parameters "stop parameter" (%), "endtime" (s), and "inflow direction" (°) had to be set in order to run RAMMS-DF. The "stop parameter" is based on the moving momentum (as a product of mass and velocity) and decides when a simulation run has to be stopped. Here, an initial threshold of 5% for the percentage of moving momentum was chosen, as stop parameters between 5 and 10% provide sufficiently good results (Bartelt et al., 2017). The "endtime" describes the time after which a simulation run is being aborted if the "stop parameter" has not yet been reached. With 4000 s, it was set rather high here so that simulations were always aborted by the "stop parameter". The "inflow direction" indicates the flow direction of the torrent (counterclockwise from the x-coordinate of the topographic data) at the location where the input hydrograph is placed (here: 100°). It is required for the first calculation step in the simulation program.



**Table 9.** Derivation of different parameters for the 3-point input hydrograph used to calibrate RAMMS-DF to the summer 2005 debris flow event.

| Parameter | Derivation/Assumptions |
|---|---|
| $V$ (m$^3$) | In the best possible case, the total released debris flow volume ($V$) is assessed by experts shortly after an event, and therefore available through reports or event documentation. If not, it can be estimated iteratively by means of various simulation runs with different volumes but constant initial values for $\mu$ and $\xi$, and validated against known (mapped) debris flow extents. The latter was carried out for this study. |
| $Q$ (m$^3$/s) | $Q_{\max} = 0.5 * (0.1 * (V^{0.833}))$ (after Rickenmann, 1999)<br>To determine the debris flow peak discharge ($Q_{\max}$), the empirical relation already implemented in RAMMS-DF between $Q_{\max}$ and $V$ from Rickenmann (1999) is used. Since this relation is based on extreme events, its results should rather be interpreted as an upper limit for $Q_{\max}$. Therefore, resulting $Q_{\max}$ values were multiplied by a correction factor of 0.5 here. |
| $t_1$ (s) | $Q_{\max}$ is expected fairly quickly after 10 seconds due to the steep terrain. |
| $v$ (m/s) | After starting the simulation in RAMMS-DF, the debris flow velocity ($v$) adapts to changes in slope defined by the input DEM. Therefore, the choice of $v$ is not of great importance for the simulation result. Considering the assumed grain size distribution and known slope, an initial debris flow velocity of 9 m/s was assumed here. |
| $t_2$ (s) | Automatically results from $V$, $Q_{\max}$ and $t_1$. |

### 4.1.6 Reconstructing the summer 2005 event sequence

Reconstruction of the exact event sequence from all available data and documentation of the summer 2005 event (chapter 2.3) was challenging. It is not known if the debris flow event that caused the sediment deposits at Creux du Pillon consisted of one single large surge or of multiple consecutive surges, nor if it occurred all in one day or in two main events (June 24 and July 29). Therefore, a most plausible event sequence was iteratively derived based on numerous model runs with RAMMS-DF. The results of these simulations showed that it is very unlikely that one single debris flow surge could have produced a similar

extent of debris flow deposits in the accumulation area as the mapped 2005 extent. It is most likely that the debris flow occurred in the form of two consecutive surges. The first surge was stopped at the hiking trail bridge (at CSP4, 1346 m a.s.l., Fig. 7a,c,d), led to deposits in the channel and defined the maximum runout of the debris flow event. Due to the impact of the first surge (tearing open the bed in the upslope sections of the channel, Fig. 7b) and its deposition (completely blocking the channel cross section along the deposited material of the first surge), the friction for the second surge was increased and the channel slope

along the accumulation area decreased. In consequence, the second surge most likely had a limited runout, broke out of the channel and led to deposits outside the channel (maximum width at CSP3, Fig. 7a).





**Table 10.** Debris flow hydrograph parameters of the reconstructed 2005 event assumed to have occurred in two surges.

|  | $V$ (m$^3$) | $Q_{max}$ (m$^3$/s) | $t_1$ (s) | $v$ (m/s) | $t_2$ (s) |
|---|---|---|---|---|---|
| 1. surge | 20'000 | 191 | 10 | 9 | 209 |
| 2. surge | 10'000 | 107 | 10 | 9 | 186 |

**Table 11.** "Best-fit" friction parameters resulting from calibration of RAMMS-DF to the 2005 event.

| Friction parameter | 1. surge (20'000 m$^3$) | 2. surge (10'000 m$^3$) |
|---|---|---|
| Dry-Coulomb friction $\mu$ (-) | 0.09 | 0.12 |
| Viscous-turbulent friction $\xi$ (m/s$^2$) | 550 | 550 |

Based on the reconstructed event sequence and the iteratively determined total debris flow volume released in the initiation zone, the final input hydrographs were defined for the 2005 event (Table 10) and could then be used for the actual calibration of RAMMS-DF to the Dar catchment (to find the "best-fit" friction parameters, chapter 4.1.7). Since the first surge of a debris

flow event is often the largest (Zanuttigh and Lamberti, 2007), the total released volume estimated to 30'000 m$^3$ (chapter 4.1.4) was divided into two-thirds (20'000 m$^3$) for the first surge and one-third (10'000 m$^3$) for the second surge.

### 4.1.7  "Best-fit" friction parameters

Finally, numerous simulation runs with RAMMS-DF applying stepwise adjusted initial values for $\mu$ and $\xi$ were necessary to best possibly reproduce the 2005 event. Only one friction parameter was changed at a time in order to clearly assign its effect

on the simulation results. First, the plausible range of the "best-fit" friction parameters was found by iteratively adapting $\mu$ by $\pm 0.01$ and $\xi$ by $\pm 100$ m/s$^2$ (from the initial values $\mu = 0.08$ and $\xi = 200$ m/s$^2$, chapter 4.1.3). Then, $\mu$ was changed stepwise by $\pm 0.005$ and $\xi$ by $\pm 10$ m/s$^2$ for fine-tuning. The "best-fit" friction parameters resulting from the model calibration are shown in Table 11. Based on the modelled extent and the impact of the first debris flow surge in the accumulation area (chapter 4.1.6), a slightly higher $\mu$ value was defined for the second surge. The composition of the released debris flow volume (proportion of

water vs. proportion and grain size distribution of loose material) and thus the final $\xi$ value was however assumed the same for both surges.



## 4.2 Resulting simulations of PPDF scenarios for the Dar catchment

After the calibration of RAMMS-DF to the Dar catchment (chapter 4.1), the potential PPDF scenarios (chapter 3.3) could be simulated. Unlike for the calibration of the model, a more recent (2016) high-resolution (0.5 m) swissALTI$^{3D}$ DEM, repre-

senting the current topography of the channel and adjacent areas, was used as input DEM to simulate the potential scenarios (Table 8). For areas below 2000 m a.s.l., the DEM was created using ALS data and has a vertical accuracy of $\pm$0.5 m. For areas above 2000 m a.s.l., the DEM was created by stereocorrelation of high-resolution (25 cm) SWISSIMAGE Level 2 orthophotos and is of lower vertical accuracy ($\pm$1 to 3 m on average, swisstopo, 2018).

Prior to the simulation of the potential PPDF scenarios, input hydrographs had to be defined for each scenario or respectively

each potential event volume (DF$_{small}$ to DF$_{very\ large}$ in Table 6 and Table 7). Definition of scenario-based debris flow event sequences and derivation of hydrographs for runout modelling is, however, not straightforward, as for a specific potential event volume numerous event sequences (for instance variable numbers of surges) are plausible. According to the reconstruction of the event sequence of the summer 2005 debris flow (chapter 4.1.6), two different debris flow event sequences (sub-scenarios) and hydrographs were defined for each PPDF scenario, one with a single surge, and one with two individual debris flow surges

(sub-scenarios DF$_1$-DF$_8$ in Table 12). For the latter, two-thirds of the total event volumes were assigned to the first surge, and one-third to the second surge. It is argued that for hazard assessment, subdividing the potential event volumes into more than two or three surges does not provide added value, as runout distance and extent of debris flow deposits decrease with smaller debris flow volumes (Rickenmann, 1999).

The final eight sub-scenarios for potential PPDFs in the Dar, i.e. four different event volumes (DF$_{small}$ to DF$_{very\ large}$) and two

event sequences per event volume, were then simulated with RAMMS-DF. The 3-point input hydrographs required to run the model were derived for each sub-scenario applying the same empirical relation between surge volume (water-debris mixture) and maximum debris flow discharge as described in chapter 4.1.4 (Table 9, Table 12, Bartelt et al., 2017). Like for the model calibration, the starting point for debris flow scenario simulation (i.e. location of the input hydrographs) was set at 1795 m a.s.l. in the channel of the Dar (light blue triangle in Fig. 1a). Following the derivation of the "best-fit" friction parameters from

model calibration (chapter 4.1.7), slightly different values for dry friction ($\mu$) but constant values for turbulent friction ($\xi$) were assigned to the different surges of the sub-scenarios (Table 12).

### 4.2.1 Intensities and runout of modelled debris flows

Maximum intensities (flow heights and flow velocities) of modelled potential debris flows for the Dar at the location of four cross sections along the channel (CSP1 to CSP4, Fig. 7a) are listed in Table 13. In general, the maximum modelled debris flow

intensities increase with larger potential event volumes or larger volumes of individual debris flow surges, respectively. The flow velocities in the steep transit zone (CSP1 just above and CSP2 just below Cascade du Dar) are always higher than in the flatter accumulation area (CSP3, Creux du Pillon) and reach at maximum 12.7 m/s at CSP1 (scenario DF$_8$). The maximum flow heights of the simulated debris flow scenarios range between 3.9 m (DF$_2$) and 9.0 m (DF$_7$).





**Table 12.** Final debris flow scenarios, sub-scenarios and parameters of input hydrographs used for numerical modelling of potential PPDFs in the Dar with RAMMS-DF.

| Potential event volume | | | Parameters of input hydrographs in RAMMS-DF | | | | |
|---|---|---|---|---|---|---|---|
| Scenario | Volume ($m^3$) | Sub-scenario | Number of surges | Volume per surge ($m^3$) | $\mu$ | $\xi$ | $Q_{max}$ ($m^3$/s) |
| $DF_{small}$ | 55'000 | $DF_1$ | 1 | 1 * 55'000 | 0.09 | 550 | 444 |
| | | $DF_2$ | 2 | 1 * 35'000 | 0.09 | 550 | 305 |
| | | | | 1 * 20'000 | 0.12 | 550 | 191 |
| $DF_{medium}$ | 120'000 | $DF_3$ | 1 | 1 * 120'000 | 0.09 | 550 | 851 |
| | | $DF_4$ | 2 | 1 * 80'000 | 0.09 | 550 | 607 |
| | | | | 1 * 40'000 | 0.12 | 550 | 340 |
| $DF_{large}$ | 190'000 | $DF_5$ | 1 | 1 * 190'000 | 0.09 | 550 | 900 |
| | | $DF_6$ | 2 | 1 * 120'000 | 0.09 | 550 | 851 |
| | | | | 1 * 70'000 | 0.12 | 550 | 543 |
| $DF_{very\ large}$ | 450'000 | $DF_7$ | 1 | 1 * 450'000 | 0.09 | 550 | 900 |
| | | $DF_8$ | 2 | 1 * 300'000 | 0.09 | 550 | 900 |
| | | | | 1 * 150'000 | 0.12 | 550 | 900 |

Runout distances for the reconstructed summer 2005 event ($DF_{c\_2005}$) and the different sub-scenarios for potential debris
flows in the Dar ($DF_1$-$DF_8$) are shown in Table 14. Runout values are calculated as the air distance from the point where
deposition begins (at 1412 m a.s.l., chapter 4.1.1, Fig. 7a) until the end of the modelled deposits. According to the modelling
results, despite the larger volumes, debris flow sub-scenarios $DF_1$ to $DF_4$ and $DF_6$ show shorter runout than the 2005 debris
flows. Only sub-scenarios with debris flow surge volumes of 190'000 $m^3$ or higher show longer runout than the 2005 debris
flows.

**4.2.2   Extents and thickness of modelled debris flow deposits**

For all PPDF sub-scenarios for the Dar ($DF_1$-$DF_8$), Fig. 8 shows the spatial extents and thickness distribution of debris flow
deposits resulting from the simulations with RAMMS-DF. The thickness of the modelled debris flow deposits corresponds
to the simulated flow heights (as a proxy for accumulation depth) at the time of reach of the defined threshold of moving
momentum (i.e. when the simulations were stopped, chapter 4.1.5 ). For debris flow sub-scenarios $DF_1$ and $DF_2$ characterized
by a potential event volume of 55'000 $m^3$, the simulated spatial extents of debris flow deposits are comparable to the 2005 event,
though with somewhat shorter runout (Fig. 8a,b). However, while debris flow deposits of sub-scenario $DF_1$ are concentrated
primarily in the lower and middle parts of the main deposition area at Creux du Pillon, the deposits associated with sub-scenario





**Table 13.** Maximum flow heights and flow velocities of modelled potential PPDFs for the Dar at the location of four cross sections along the channel (see Fig. 7a). Values are listed for the calibrated model runs of the reconstructed 2005 event ($DF_{c\_2005}$) and for the eight sub-scenarios ($DF_1$-$DF_8$).

| | Simulation | | CSP1 | | CSP2 | | CSP3 | | CSP4 | |
|---|---|---|---|---|---|---|---|---|---|---|
| Sub-scenario | Surge | Volume (m$^3$) | Max. flow height (m) | Max. flow velocity (m/s) | Max. flow height (m) | Max. flow velocity (m/s) | Max. flow height (m) | Max. flow velocity (m/s) | Max. flow height (m) | Max. flow velocity (m/s) |
| $DF_{c\_2005}$ | 1 | 20'000 | 3.1 | 8.6 | 2.6 | 9.7 | 2.6 | 5.7 | 0.5 | 2.0 |
| | 2 | 10'000 | 2.6 | 6.4 | 1.9 | 8.6 | 2.4 | 2.4 | - | - |
| $DF_1$ | - | 55'000 | 4.8 | 10.1 | 4.1 | 11.3 | 4.6 | 6.0 | - | - |
| $DF_2$ | 1 | 35'000 | 4.3 | 9.5 | 3.9 | 10.0 | 3.9 | 5.4 | - | - |
| | 2 | 20'000 | 3.4 | 8.3 | 1.9 | 7.0 | 1.5 | 1.3 | - | - |
| $DF_3$ | - | 120'000 | 6.1 | 11.7 | 4.0 | 10.8 | 5.9 | 7.1 | - | - |
| $DF_4$ | 1 | 80'000 | 5.5 | 10.8 | 5.5 | 11.8 | 5.1 | 6.6 | - | - |
| | 2 | 40'000 | 4.5 | 9.0 | 2.4 | 8.9 | 1.6 | 0.9 | - | - |
| $DF_5$ | - | 190'000 | 6.2 | 11.8 | 6.7 | 11.2 | 5.7 | 7.2 | 2.3 | 5.8 |
| $DF_6$ | 1 | 120'000 | 5.0 | 11.8 | 4.4 | 10.6 | 5.8 | 7.2 | - | - |
| | 2 | 70'000 | 5.0 | 10.3 | 4.1 | 11.2 | 2.0 | 1.5 | - | - |
| $DF_7$ | - | 450'000 | 6.1 | 12.0 | 6.4 | 11.3 | 9.0 | 7.3 | 3.6 | 7.2 |
| $DF_8$ | 1 | 300'000 | 6.2 | 12.7 | 6.4 | 11.2 | 7.7 | 7.4 | 3.4 | 5.6 |
| | 2 | 150'000 | 6.2 | 11.7 | 4.2 | 10.7 | 7.9 | 5.7 | - | - |

$DF_2$ (two debris flow surges) show a slightly more widespread picture and a stronger concentration of accumulated material in the middle and upper parts of the deposition area. Maximum thickness of debris flow deposits is 4.3 m for sub-scenario

$DF_1$, and 5.1 m for the sum of the two debris flow surges of sub-scenario $DF_2$ ($DF_2$ surge 1: max. 3.6 m, surge 2: max. 3.2 m; Fig. 8a,b).

Debris flow sub-scenarios with a potential event volume of 120'000 m$^3$ ($DF_3$ and $DF_4$) still show shorter runout than the 2005 event (Fig. 8c,d). The spatial debris deposition patterns are similar to those just described for sub-scenarios $DF_1$ and $DF_2$, with the exception that the area of accumulated loose material as well as zones with thick debris deposits (>3 m) are larger and

the average thickness of deposits is higher. Maximum thickness of debris flow deposits amounts to 6.7 m for sub-scenario $DF_3$,





**Table 14.** Runout of modelled potential PPDFs for the Dar. Values are listed for the calibrated model runs of the reconstructed 2005 event (DF$_{c\_2005}$) and for the eight sub-scenarios (DF$_1$-DF$_8$).

| | **Simulation** | | | **Runout** |
|---|---|---|---|---|
| Sub-scenario | Surge | Volume (m$^3$) | Elevation at end of modelled debris flow deposits (m a.s.l.) | Air distance (m) from the beginning of the 2005 debris flow deposits (at 1412 m a.s.l., Fig. 7a) until the end of modelled deposits |
| DF$_{c\_2005}$ | 1 | 20'000 | 1346 | 736 |
| | 2 | 10'000 | 1379 | 315 |
| DF$_1$ | - | 55'000 | 1354 | 633 |
| DF$_2$ | 1 | 35'000 | 1356 | 593 |
| | 2 | 20'000 | 1386 | 242 |
| DF$_3$ | - | 120'000 | 1349 | 713 |
| DF$_4$ | 1 | 80'000 | 1351 | 677 |
| | 2 | 40'000 | 1386 | 245 |
| DF$_5$ | - | 190'000 | 1182 | 2545 |
| DF$_6$ | 1 | 120'000 | 1349 | 713 |
| | 2 | 70'000 | 1385 | 258 |
| DF$_7$ | - | 450'000 | 1168 | 2838 |
| DF$_8$ | 1 | 300'000 | 1172 | 2722 |
| | 2 | 150'000 | 1368 | 418 |

and up to 6.4 m in total for the sum of both debris flow surges of sub-scenario DF$_4$ (DF$_4$ surge 1: max. 5.3 m, surge 2: max. 4.2 m; Fig. 8c,d).

A clearly different picture emerges from the comparison of sub-scenarios DF$_5$ and DF$_6$ (Fig. 8f vs. Fig. 8e). For these two scenarios, the defined number of debris flow surges remarkably influences the modelled runout, extent and thickness of deposition. In sub-scenario DF$_5$, the defined total event volume of 190'000 m$^3$ is released as one single debris flow surge and the modelled runout shows that debris flow deposits would reach the edge of Les Diablerets village. The majority of transported loose material would transit through the main accumulation area of the 2005 debris flow and be deposited further downstream near the village (Fig. 8f). For sub-scenario DF$_5$, damage to infrastructure and buildings in Les Diablerets caused by the transported debris can thus not be excluded. If the potential event volume of 190'000 m$^3$ is partitioned into a first surge of 120'000 m$^3$ and a second surge of 70'000 m$^3$ (as in sub-scenario DF$_6$), the debris flow runout distance is significantly





**Figure 8.** (a-e) Modelled runout, spatial extents and thickness distribution of debris flow deposits for the PPDF sub-scenarios DF$_1$ to DF$_4$ and DF$_6$ for the Dar (Tables 13 and 14) with deposition area at Creux du Pillon (Fig. 1a,c and Fig. 3) and maximum runout shorter than the summer 2005 debris flow event. (f-h) Modelled runout, spatial extents and thickness distribution of debris flow deposits for the PPDF sub-scenarios DF$_5$, DF$_7$ and DF$_8$ for the Dar (Tables 13 and 14) with deposition area at Creux du Pillon and Les Diablerets (Fig. 1a,c and Fig. 3) and maximum runout significantly longer than the summer 2005 debris flow event. Underlain in (a)-(h) is a hillshade of the 2016 swissALTI$^{3D}$ DEM (2 m resolution) from swisstopo.




shorter and its extent significantly smaller compared to sub-scenario DF$_5$. For sub-scenario DF$_6$, the debris flow deposits remain confined to the main deposition area of the 2005 event (Creux du Pillon). In this case, no damage caused by debris flows is to be expected in Les Diablerets.

Modelled PPDFs with a defined total event volume of 450'000 m$^3$ (sub-scenarios DF$_7$ and DF$_8$) show the longest runout
and the highest thickness of accumulated loose material (Fig. 8g,h). For sub-scenario DF$_7$, the majority of transported loose material is deposited close to or within the village of Les Diablerets. If two debris flow surges occur with the same total volume (sub-scenario DF$_8$), deposits of the first surge would be concentrated at the edge of the village, whereas the second surge would lead to a strong accumulation of loose material upstream, within the main deposition area of the 2005 event. Maximum thickness of modelled debris flow deposits is reached for the sub-scenario DF$_8$ with up to 11.4 m for the sum of both debris
flow surges (DF$_8$ surge 1: max. 9.3 m, surge 2: max. 11.4 m; Fig. 8h). Independent of the number of surges (either one (DF$_7$) or two (DF$_8$) debris flow surges), PPDFs with a total event volume of 450'000 m$^3$ would in both cases reach the village of Les Diablerets, where damage to infrastructure and buildings can be expected.

## 5 Discussion

### 5.1 Scenario building for PPDFs

The presented multi-methods approach for debris flow scenario building is specifically designed for small pro- and periglacial catchments with scarce past event data. The detailed assessment of precipitation-related triggering events in combination with calculations of clear water discharges per event allows the definition of realistic triggering scenarios and the assignment of estimated return periods to potential debris flow volumes. The latter result from both empirical approaches and detailed field investigations. Despite the various shortcomings and uncertainties that are further discussed below, the proposed approach
allows the definition of sound debris flow scenarios and can easily be applied for other alpine torrential catchments, though requiring sometimes some adjustment depending on data availability, catchment accessibility and available time for debris flow hazard assessment. To our knowledge, studies including this level of detail are rare, despite the obvious need for sound scenario building for debris flow hazards in pro-/periglacial catchments under climate change.

### 5.1.1 Potential PPDF initiation zones

Prior to scenario building, potential debris flow initiation zones have to be identified (Fig. 4). For unstudied catchments, this can be done by the creation of susceptibility maps (e.g., Lancaster et al., 2012), best in combination with verification and independent assessment in the field (cf. Rickenmann and Zimmermann, 1993). Here, we thoroughly studied potential debris flow initiation zones in the cirque of Le Dar Dessus (chapter 2.2, Fig. 2, Appendix D). Large parts of the cirque are steep enough for debris flow initiation and composed of supply-unlimited loose material (cf. Sattler, 2016). The eastern parts of
the cirque are even continuously replenished through gravitational processes with sediments originating from above the first steep rock step (Fig. 2a). Debris flow initiation is thought to occur mainly by liquefaction of water-oversaturated sediments



in or along one (or several) of the numerous channels cutting through the sediment deposits in the cirque of Le Dar Dessus. Progressive erosion at the contact zone of the rock threshold constituting the first steep rock step and the uppermost parts of the sediment deposits in the cirque may also lead to debris flow triggering (Fig. 2a, Fig. A1b, Fig. D2e). Even if Lambiel et al. (2008) assume that permafrost degradation (active layer thickening) might have facilitated debris flow initiation during the summer 2005 event, the spatial distribution, internal structure and thermal state of permafrost in the cirque of Le Dar Dessus still needs to be investigated in more detail in order to better understand its potential influence on debris flow initiation.

### 5.1.2 Definition of meteorological and hydrological triggering scenarios

The proposed methodological approach for scenario building and runout modelling of PPDFs (Fig. 4) is also applicable without the definition of meteorological and hydrological triggering scenarios. Neither the required steps to define bed load scenarios and total event volumes per scenario (chapters 3.2 and 3.3), nor the simulation of the different scenarios with RAMMS-DF (chapter 4) directly rely on precipitation or runoff scenarios. We argue that this is an advantage of our proposed approach because data on extreme point rainfalls with varying return periods needed to define possible meteorological triggering scenarios specifically for the debris flow initiation zones (chapter 3.1) might not always be available (especially not with a ground resolution of 1 km$^2$). Leaving out the definition of meteorological and hydrological triggering scenarios can be considered a shortcut within our proposed workflow in case data availability and/or time for hazard assessment is limited.

However, we want to emphasize the benefits of defining, if feasible, possible meteorological and hydrological triggering scenarios according to our proposed approach (chapter 3.1). Together with resulting clear water discharges (peak values and duration, Fig. 6) for thunderstorm, long-lasting rainfall and ROS scenarios computed with a precipitation-runoff model (here: ZEMOKOST), it is possible to assign plausible triggering events and triggering mechanisms to the different debris flow scenarios (total event volumes). In addition, it allows the definition of probabilities of occurrence for debris flow scenarios, even for catchments with no or only scarce past event data (Table 7). This is very useful for hazard assessment, as the creation of hazard matrices and hazard maps requires this type of information (e.g., Hürlimann et al., 2006; Frey et al., 2018). Moreover, the meteorological and hydrological triggering scenarios can also be used for hazard assessment of flooding and (as opposed to debris flows) other types of bed load transport potentially occurring in the torrent catchment.

Precipitation scenarios for the cirque of Le Dar Dessus are derived from data representing levels of rare heavy precipitation events in today's climate (chapter 3.1.1, Frei and Fukutome, 2022). This might be judged as a shortcoming because debris flow triggering meteorological events are expected to occur more frequently and more intensely in future with ongoing climate change (Gobiet et al., 2014; Hirschberg et al., 2021). However, due to the rapid and dynamic changes in the debris flow initiation zones (e.g., Beniston et al., 2018), PPDF scenarios should in any case only be established for a relatively short time period (scenarios' relevance of ~20 to 30 years, Allen et al., 2022). After every major event or at latest after exceedance of the scenarios' relevance, scenarios need to be re-evaluated and hazard assessment repeated. Therefore, it is probably better to work with existing data on current extreme point precipitation instead of working with future climate projections, for which uncertainty increases with time.



Our proposed approach to define precipitation scenarios for thunderstorms and long-lasting rainfalls (chapter 3.1.1) represents a simplification of reality. For thunderstorm scenarios, hourly precipitation sums (mm) are linearly interpolated to five minutes intervals and reach maximum intensities after 30 minutes, while for long-lasting rainfall scenarios, daily precipitation sums are evenly distributed over 24 h, so that constant intensities prevail (Table 1). For the same respective event sizes, the duration and peak intensities of rainfall events that might occur in reality may thus differ significantly.

Concerning ROS events, the involved processes are more complex and highly variable over small spatiotemporal scales (e.g., Würzer, 2018). They are, however, relevant for the triggering of debris flows in pro- and periglacial catchments (Rössler et al., 2014), hence our attempt to include them in our scenarios. One of the most important challenges in building ROS scenarios relates to the availability of distributed data on snow depth, snowpack characteristics and SWE of the end-of-winter snow cover for the debris flow initiation zones. If no estimates can be made from available in-situ observations (as we do

here), using remotely-sensed snow cover and SWE products (e.g., Naegeli et al., 2022; Tsang et al., 2022) might be an option, even if the spatial resolution of these data sets might still be too low for analyses at the local scale. Here, several simplifying assumptions were made for the definition of ROS scenarios. They are mostly related to the (i) duration of rainwater absorption by the snowpack, (ii) defined quotients to calculate snow melt intensities from rainfall intensities, (iii) assumed value and spatial distribution of SWE, and (iv) assumed homogeneous characteristics of the snowpack. Moreover, the shortcomings of

our long-lasting rainfall scenarios discussed above obviously hold also true for our ROS scenarios. However, despite all these assumptions and simplifications, we argue that our approach to build scenarios for ROS events (chapter 3.1.2) is meaningful, straightforward and rather easy to apply. Even if simplifying, it is, to our knowledge, among the first attempts to systematically build scenarios for ROS events within PPDF hazard assessment.

The different precipitation scenarios developed here do not account for weather events that might occur shortly to a couple of

days prior to the potentially debris flow triggering meteorological event. According to our approach, it is therefore not possible to account for variable hydro-geological pre-event conditions (varying water content in the substratum). Being part of the variable disposition for debris flows, these hydro-geological pre-event conditions are relevant as they influence the triggering mechanisms and magnitude of an event (Zimmermann et al., 1997a). However, we argue that the broad range of meteorological and hydrological triggering scenarios (chapter 3.1, Fig. 6) as well as bed load scenarios (Table 6) defined for the cirque of Le

Dar Dessus integrates, at least to some extent, the possible effects of variable hydro-geological pre-event conditions on the triggering and volume of debris flows.

In addition, precipitation independent events may also lead to the triggering of PPDFs (e.g., Chiarle et al., 2007, chapter 3.1). In similar geomorphological context compared to the cirque of Le Dar Dessus, it is for instance not rare for debris flows to be triggered by snow melt only (e.g., Decaulne and Sæmundsson, 2007; Kummert et al., 2018). Snow melt only scenarios are

not specifically defined here, but we argue that for the cirque of Le Dar Dessus, both our long-lasting rainfall and snow melt dominated ROS scenarios sufficiently represent the range of possible snow melt only scenarios. Indeed, both defined long-lasting rainfall and ROS scenarios would lead to water oversaturation of the ground in a similar way to a typical snow melt only event.





Finally, our proposed approach for PPDF scenario building (Fig. 4) does not explicitly consider cascading processes, which
are known to have a greater potential to trigger catastrophic events (e.g., Vuichard and Zimmermann, 1987). However, we
thoroughly assess the potential rock fall impact disposition from the northwestern flank of Oldehore/Becca d'Audon into the
newly forming small lake above the first steep rock step and implications of potential GLOFs for the PPDF initiation zones in
the cirque of Le Dar Dessus (Appendix B, Fig. B1-B3, Table B1). Due to the small potential outburst volume and the rather
low likelihood of such an event, GLOFs are neglected as potential triggering mechanisms for PPDFs starting in the cirque of
Le Dar Dessus. For catchments where GLOFs cannot be excluded as PPDF triggering mechanisms, we propose to carry out
scenario building according to recently established approaches (Schneider et al., 2014; Frey et al., 2018; Allen et al., 2022).

### 5.1.3    Definition of bed load scenarios

The core of our multi-methods approach for PPDF scenario building is the derivation of bed load scenarios (Fig. 4, chapter 3.2).
Based on recommendations for glacier- and permafrost-related hazard assessments (Allen et al., 2022, p. 22: *"The goal of*
*scenario development is to establish at least three feasible scenarios as input to process-based hazard models, where the*
*potential mass or volume initiated in a small, medium, or large event is estimated [. . . ]"*), we propose to build bed load
scenarios for small, medium and large events using empirical methods especially designed for small and steep high mountain
torrent catchments (Spreafico et al., 1996, chapter 3.2, Appendix D). In addition, we recommend to establish a "worst-case"
(here: "very large") bed load scenario to account for potential extreme events with very low probability of occurrence (Loat
and Petrascheck, 1997; Allen et al., 2022).

The bed load volume estimation approach developed by Spreafico et al. (1996) is based on the analysis of several hundreds
of debris flows triggered during the 1987 heavy precipitation event in the Swiss Alps. As more than 50% of the recorded debris
flows had originated in proglacial areas in 1987 (Zimmermann and Haeberli, 1992; Rickenmann and Zimmermann, 1993), we
argue that the approach is especially well suited to build bed load scenarios for PPDFs. Bed load volume estimates based on
Spreafico et al. (1996) require expert knowledge, high-resolution DEMs and orthophotos. They can theoretically be carried
out from remote, but we highly recommend a thorough field-based assessment and plausibilisation of the resulting bed load
volumes.

For that purpose, the estimates of bed load volumes for different event sizes based on Spreafico et al. (1996) can be com-
plemented with and validated against standardized field-based methods such as SEDEX$^{©}$ (Frick et al., 2011, chapter 3.2,
Appendix D). Other approaches used in practice exist (e.g., Zimmermann et al., 1997b; Angerer et al., 2003; Gertsch, 2009),
but their suitability for bed load volume estimates specifically for PPDFs was not tested here. In this context, recent advances
in close-range remote sensing techniques seem to open promising possibilities for automated derivation of geometrical param-
eters continuously along the torrent channel (Schmucki et al., 2023). Applied to initiation zones of PPDFs, resulting data sets
could be used for possibly more accurate estimates of bed load volumes and more efficient hazard assessment compared to
commonly used approaches.

For the cirque of Le Dar Dessus, resulting bed load volumes derived either following Spreafico et al. (1996) or with SEDEX$^{©}$
are in the same order of magnitude (Table 5). In addition, the range of resulting bed load potentials calculated with common





empirical formulas only (cf. Rickenmann, 1999) overlaps quite well (except for the value calculated with the formula by Kronfellner-Kraus (1984)) with the range of bed load volumes for small to large events derived by the empirical and field-based
methods relying on catchment-specific assessments (Table 5). However, the resulting average and median of the different bed load potentials are much closer to the large than to the medium bed load scenario (chapter 3.2, Tables 5-6). This is in line with the fact that bed load potentials calculated using empirical formulas only are somewhat biased towards (very) large events (Rimböck et al., 2013), and resulting values are very sensitive to slight changes to the input variables of the formulas. Therefore, using the range of resulting bed load potentials calculated with empirical formulas to define different bed load
scenarios should only be considered if time for hazard assessment is very limited, catchment-specific data very scarce and field surveys impossible.

The bed load volumes defined for the Dar using our approach are either in the same order of magnitude (for the small scenario) or significantly higher than the debris flow volume of 30'000 m$^3$ reconstructed for the 2005 event (chapter 4.1.6) with an estimated return period of 50 to 100 years (chapter 2.3). In this regard, our estimated bed load volumes (Table 6) all
represent major to extreme erosional events. Therefore, we assigned rather low probabilities of occurrence to our scenarios (30 to 50 years for the "small" scenario to (highly) unlikely for the "very large" scenario, Table 7). Based on our knowledge of the geomorphological functioning of the Dar catchment (chapter 2, Appendix A), we know that smaller PPDF events with low magnitude and high frequency (return period in the order of $10^0$ years) are not relevant for hazard assessment as their maximum runout is located shortly after the second steep rock step (incised intermediate fan in Fig. A1c), far from potentially
vulnerable infrastructure.

On the other hand, we consider the very large scenario (Table 6) to be very unlikely to happen as it would require several to all channels in the cirque of Le Dar Dessus to simultaneously experience strong longitudinal (channel) and lateral (embankment) erosion (Fig. D2d,e). In such steep and widespread accumulation of loose sediments as the cirque of Le Dar Dessus (estimated total volume of loose material in the order of few $10^6$ m$^3$), erosion occurs mainly linearly along channels, and it is rare that
debris flow channels function all at the same time once one or few channels start to be strongly incised. In this case, water tends to concentrate in these channels, limiting the erosion in other parts of the cirque. From a geomorphological perspective, it is thus difficult to imagine a very large scenario with a bed load volume of 300'000 m$^3$ to be mobilised in one single event. Such very specific geomorphological aspects are generally poorly represented in the various approaches available to estimate bed load volumes and derive bed load scenarios. Geomorphological expertise and knowledge of the local conditions remains
essential for PPDF scenario building and hazard assessment.

### 5.1.4 Definition of final debris flow scenarios and sub-scenarios

In most cases and especially for catchments with scarce past event data, no observations or measurements are available for the volumetric solids concentration of PPDFs. The catchment- and event-specific ratio of water-debris mixture can, however, be narrowed down to a certain extent using empirical values and analogies (e.g., Iverson et al., 1997). Here, it is estimated based
on empirical considerations linking both the slope and grain size distribution of the debris flow initiation zones to the flow behaviour (cf. Takahashi, 1991; Tognacca, 1999; Weber and Rickenmann, 2000, chapter 3.3). Total event volumes (volumes of



water-debris-mixture released during an event, i.e. the final debris flow scenarios), can then be calculated by multiplication of bed load volumes for different scenarios with this estimated ratio of water-debris mixture reflecting catchment characteristics (1.5 for the cirque of Le Dar Dessus, chapter 3.3, Table 6). For simplicity and due to the lack of relevant data, the ratio of
water-debris mixture is assumed constant here for all debris flow scenarios (Table 6). In reality, the proportions of water and fine to coarse-grained sediments may vary within the same catchment, and even between subsequent surges of a single debris flow event (e.g., Zanuttigh and Lamberti, 2007).

As there are typically scarce or no data on past events, the definition of sub-scenarios from PPDF scenarios and their total event volumes (i.e. definition of potential triggering conditions, event sequences, number of surges, hydrograph per surge,
flow characteristics) is inherently challenging. For the cirque of Le Dar Dessus, the plausible event sequence reconstructed for the to date only recorded major debris flow event of summer 2005 (with two distinct surges, chapter 4.1.6) is used to define sub-scenarios (Table 12), assuming that future debris flows would behave similarly. In the total absence of records on past events, analogies from well-studied PPDF events occurring in catchments with similar geomorphological characteristics could be used. In addition, any future PPDF event should be thoroughly analysed, leading to a better process understanding of the
studied catchment. This would allow scenarios and sub-scenarios to be defined more realistically and to be gradually refined.

## 5.2 Runout simulations of PPDF scenarios with RAMMS-DF

Here we use RAMMS-DF (Bartelt et al., 2017) for numerical runout modelling of all our PPDF (sub-)scenarios for the Dar (Fig. 4, chapter 4). Of course, many other (commercial or freely available) debris flow simulation approaches exist, developed by research institutions or the private sector (e.g., FLOW-2D, LAHARZ, MASSMov2D, RAMMS-DF, TopRun-DF/TopFlow
DF, Flow-R for a non-exhaustive list of two-dimensional debris flow models).

If records of past events exist, RAMMS-DF can be calibrated to the studied pro-/periglacial catchment. For that purpose, a DEM representing channel topography prior to the event must be used. PPDF scenarios should then be simulated using the most recent DEM. Fortunately, high-resolution DEMs exist for the Dar for both just before the summer 2005 event and for today (Table 8). For data scarce areas, the availability of reliable DEMs may pose a problem, especially if the only past event
against which the model is to be calibrated dates back several decades.

The work steps followed to simulate our PPDF scenarios using RAMMS-DF, including the calibration of the model to the only known major event recorded for the Dar, are described in detail in chapter 4.1. We argue that they can easily be applied for simulations of PPDFs in other catchments, though calibration is only possible with data from at least one past event. In the absence of such data, a range of values can still be estimated for the friction parameters (Fig. 9).
The initial values for $\mu$ (dry friction) can be reconstructed from the mean channel slope in the accumulation area of debris flow deposits (Bartelt et al., 2017). For $\xi$ (turbulent friction), they are best estimated using proxy data for flow behaviour, i.e. through geomorphological, topographical and hydro-meteorological considerations (grain size distribution of loose material in the debris flow initiation zone, slope of the torrent channel and triggering precipitation scenarios). Like this, the calibration parameters $\mu$ and $\xi$ for RAMMS-DF can already be narrowed down (Bartelt et al., 2017). If no data on past events are available
for the studied catchment, the range of plausible friction values to be used for the actual PPDF scenario simulations remains





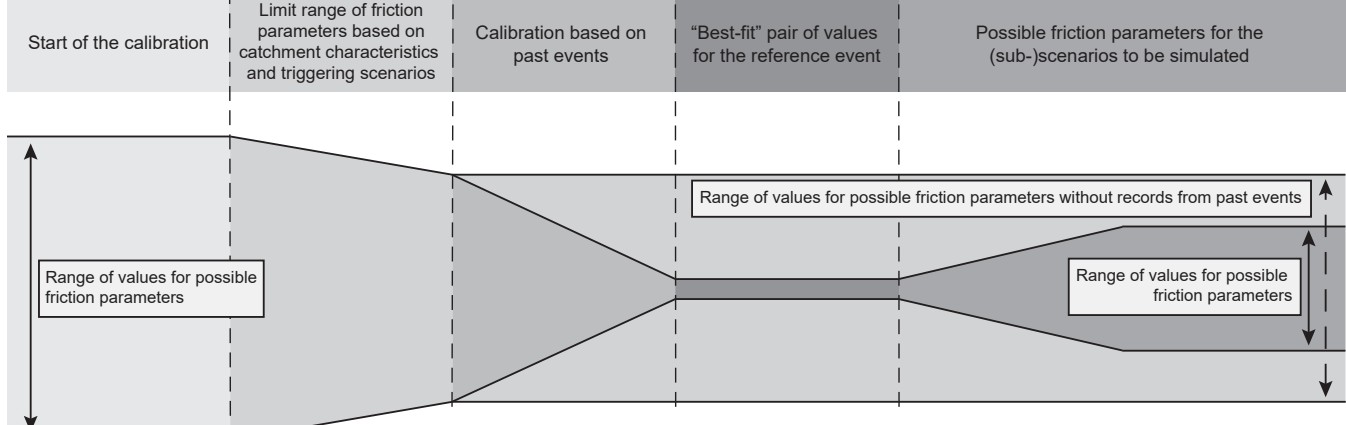

**Figure 9.** Schematic graph for the varying range of plausible friction parameters $\mu$ and $\xi$ during the calibration of RAMMS-DF to a reference event and for simulation of defined debris flow (sub-)scenarios with RAMMS-DF (Table 12 for the simulation of PPDF scenarios for the Dar). If no records of past events are available based on which RAMMS-DF can be calibrated to the studied catchment, the range of possible friction parameters $\mu$ and $\xi$ can be limited using information about geomorphological and topographical catchment characteristics and the nature of plausible triggering scenarios.

quite large (Fig. 9), increasing uncertainties related to the simulated results to an important extent. If records from past events are available, $\mu$ and $\xi$ can be stepwise adjusted (only one parameter should be changed at a time) until RAMMS-DF best possibly reproduces the documented event ("best-fit" pair of friction values, Fig. 9). The range of values for the friction parameters should then be slightly extended from the "best-fit" friction values found during the model calibration (Fig. 9)

before simulating the defined PPDF scenarios. This way, variable flow behaviour that might occur for different triggering scenarios, different scenario sizes, and subsequent surges of the very same event can be taken into account. For instance, for the simulations of our PPDF scenarios for the Dar, based on the reconstructed event of 2005, we slightly adjust $\mu$ for the second surge of sub-scenarios with multiple surges (chapter 4.1.6, Tables. 10-12).

Since recently (version 1.7.0), erosion or bed load entrainment along the flow path can also be simulated with RAMMS-DF.

For debris flows characterized by strong incision into channels and/or debris cones, the entrainment of material should not be neglected while simulating past and future events (Frank et al., 2015). However, entrainment was not accounted for in this study. In the case of the Dar catchment, field investigations showed that the volumes of sediments that could be entrained along the main channel in the transit area between the second and third steep rock step are minor if not negligible compared to the volumes of loose material released in the cirque of Le Dar Dessus (Fig. A1b,d). In addition, Condrau (2019) showed

that accounting for erosion in RAMMS-DF primarily improves the accuracy of modelled flow behaviour along the transit area, whereas neglecting erosion in RAMMS-DF results in slightly better representations of modelled extents of debris flow deposits. As debris flow extents represent one of the most important outputs in the perspective of hazard assessment, the option of including erosion and deposition along the flow path to our simulations was discarded. Moreover, including entrainment



along the channel in RAMMS-DF would probably only change the deposition characteristics of the simulated scenarios for
PPDFs travelling beyond the deposits of the summer 2005 event (i.e. large and very large scenarios). This is because additional
loose material from bed and bank erosion between Creux du Pillon and the confluence of the Dar with La Grande Eau would
probably increase the transported sediment volume to Les Diablerets (Fig. A1f,g,h).

Numerical runout modelling of debris flow scenarios is a well-established part of hazard assessment (Hürlimann et al., 2006).
However, the reliability and use of debris flow modelling, independent of which model is used, strongly varies for different
situations. For a catchment with comparatively simple terrain characteristics and many observations (i.e. many data on past
events), more realistic scenarios can be defined, and model calibration is generally of better quality. This in turn leads to more
reliable simulation results and less discretion for hazard assessment by experts. However, for a catchment with comparatively
more complex terrain characteristics and scarce to inexistent data on past events, as it may often be the case for pro- and
periglacial catchments, scenario building is challenging and prone to uncertainty (chapter 5.1). In this case, the simulated
scenarios are less reliable (at worst unrealistic) and there is more discretion for the experts in the hazard assessment (Hunziker
et al., unpublished).

If model calibration to past events is possible, we recommend assessing its quality (e.g., Beguería, 2006). If it is good, the
probability of simulated PPDF scenarios to be more realistic and reliable increases. For the Dar, achieved model performance
indices (*model efficiency*, *sensitivity* and *accuracy*) for the calibration of RAMMS-DF to the 2005 event are satisfying (Ap-
pendix E, Fig. E1). The mapped extents of the 2005 debris flow deposits are neither overestimated nor underestimated by the
calibrated model. This shows that careful calibration of RAMMS-DF based on mapped extents of only one past PPDF event
may still be a meaningful way to best possibly carry out hazard assessment for PPDF scenarios.

Even if the different work steps comprised in scenario building and runout modelling (Fig. 4) of PPDFs are carried out as
seriously as possible by experts, simulation results of PPDF scenarios should be handled with care and treated as what they
are: best possible representations of what could possibly happen in the future.

## 5.3    Implications of the presented results for debris flow hazard in the Dar catchment

Numerical simulations with RAMMS-DF for all our scenarios and sub-scenarios for the Dar catchment show that only large to
very large PPDFs would reach the village of Les Diablerets and potentially cause damage (Fig. 8, Table 14). We estimate the
probability of occurrence for such events to happen is very low (return period of several hundreds of years for the large event)
to (highly) unlikely (for the very large event, Table 7, chapter 5.1.3). Interestingly, the small and medium PPDF scenarios are
characterized by larger volumes but shorter runout distance compared to the 2005 event. This can be explained by the changes
in the channel geometry (further reduced slope angle and relatively increased friction) at Creux du Pillon as a result of the
deposits of the 2005 event, favouring sediment accumulation over transit. The threshold for potential debris flow volumes
(water-debris mixture), above which PPDFs starting in the cirque of Le Dar Dessus travel (far) beyond the extents of the
summer 2005 debris flow deposits at Creux du Pillon (Fig. 7), lies somewhere between 120'000 and 190'000 m³.

According to our scenarios, only ROS events or a very large thunderstorm event (maximum precipitation intensities of 139
mm/h, hourly precipitation sum of 72 mm, return period of 200 years, chapter 3.1.1, Table 1) could trigger a PPDF event of



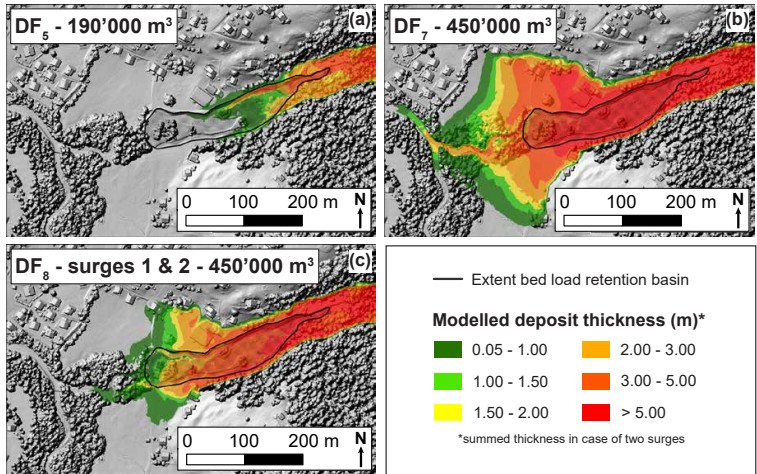

**Figure 10.** Modelled runout, spatial extents and thickness distribution of debris flow deposits for the PPDF sub-scenarios (a) $DF_5$, (b) $DF_7$ and (c) $DF_8$ (Table 14) at the location of the bed load retention basin (black-lined polygon) completed around 2017. Underlain is a hillshade of the 2018 swissSURFACE³D DEM (dm resolution) from swisstopo.

190'000 m³ in the cirque of Le Dar Dessus (Table 7). Moreover, only if PPDFs of this size would flow down as one single surge they would reach the bed load retention basin close to the confluence of the Dar with La Grande Eau, but hardly cause any
damage to infrastructure (Fig. 8f, Fig. 10a, Fig. A1i). Only for the very large PPDF scenario of 450'000 m³ (extreme event), debris flows would have the potential to break out of the retention basin, and cause damage to several buildings (Fig. 10b,c).

Hence, to potentially endanger infrastructure and lives at Les Diablerets, the event volume of PPDFs would need to be about one order of magnitude larger compared to the 2005 event (chapter 4.1.6). Such a scenario is very unlikely to happen both for (i) meteorological reasons (triggered only by snow melt dominated ROS events (Table 7) hardly ever documented for the Swiss
Alps (Würzer and Jonas (2017)) and (ii) geomorphological reasons (release of a very large bed load volume of 300'000 m³ in the cirque of Le Dar Dessus in one single event is rather implausible, chapter 5.1.3).

However, hazards for the lower parts of the Dar catchment directly related to PPDFs (flooding, high bed load transport and potentially process interactions) have to be reckoned with. According to our scenarios, small to medium debris flow events (estimated return periods of 30 to 50 respectively 100 years, Table 7) would be stopped at Creux du Pillon due to the abrupt
change in slope of the Dar after the third steep rock step (Cascade du Dar). Comparable to the 2005 event, subsequent flooding further downvalley could still occur, leading to bed and bank erosion along the more gently inclined sections of the torrent (Appendix A, Fig. A1g). Hence, reloading of the floods with important volumes of bed load material could pose certain problems to the village of Les Diablerets (chapter 2.3, Fig. 3). Moreover, the reactivation of landslides along a tributary torrent of the Dar (Ruisseau des Roseyres, Fig. A1, Jaboyedoff et al., 2009) in the course of heavy precipitation events could transport
further loose material to the lowermost parts of the Dar (B+C Ingénieurs SA, 2013).



## 6  Conclusions

In high mountain areas, climate change, rapid glacier retreat and permafrost degradation is known to increase both the disposition, probability of occurrence and potential magnitude of pro- and periglacial debris flows (PPDFs). Because of their comparatively long runout distance and potential destructive power, PPDFs can represent important threats to the downstream
areas, infrastructure and settlements.

For debris flow hazard assessment at local scale, it is common to apply multi-methods and multidisciplinary approaches including (i) geomorphologic and geologic analysis as well as scenario building, (ii) runout analysis, (iii) hazard zonation, and (iv) hazard mitigation (e.g., Hürlimann et al., 2006). Scenario building is a delicate but crucial part of the hazard assessment as it strongly influences the results of the aforementioned steps (ii) to (iv).

Scenario building for PPDFs is especially challenging as typically only scarce or no data on past events are available (Chiarle et al., 2007; Frey et al., 2016). This prevents the reconstruction of magnitude-frequency relationships (Zimmermann et al., 1997a) based on which debris flow scenarios are commonly derived (Jakob et al., 2016). Systematic scenario building approaches for PPDFs are few and mostly focus on debris flows triggered by GLOFs (e.g., Schneider et al., 2014; Frey et al., 2018), even if PPDFs are also often triggered by heavy precipitation events or/in combination with intense snow melt (e.g.,
Rickenmann and Zimmermann, 1993).

In this article, we presented a multi-methods approach for debris flow hazard scenario building that that does not rely on comprehensive past events documentation and that is especially designed for pro-/periglacial debris flows triggered by precipitation events (thunderstorms, long-lasting rainfalls, ROS events). The developed approach includes (i) the definition of precipitation dependent debris flow triggering scenarios and the calculation of associated runoff, (ii) the estimation of bed
load volumes potentially triggered per scenario, (iii) the assignation of total event volumes (water-debris-mixtures) to each triggering scenario and (iv) the definition of debris flow (sub-)scenarios for individual total event volumes (event sequence, number of surges, hydrograph per surge, flow characteristics, potential triggering conditions).

We applied the proposed approach to the torrential catchment of the Dar (~10.5 km$^2$) close to Les Diablerets (western Swiss Alps), for which only one single major debris flow event starting in pro- and periglacial environments (the cirque of Le Dar
Dessus) has been recorded in summer 2005. PPDF scenarios were defined for small, medium, large and very large events, and assigned to potential precipitation dependent triggering conditions. Return periods were estimated for each scenario based on the probability of occurrence of the triggering meteorological events. They vary between 30 to 50 years for the small scenario and a (highly) unlikely occurrence for the very large (extreme) scenario.

Numerical runout modelling of our PPDF scenarios and sub-scenarios was carried out using RAMMS-DF (Bartelt et al.,
2017). We calibrated the model using mapped extents of the 2005 debris flow deposits. Our results show that it is (highly) unlikely that PPDFs starting in the cirque of Le Dar Dessus would reach the village of Les Diablerets and endanger infrastructure and lives. However, hazards for the lower parts of the Dar catchment directly related to potential PPDFs (subsequent flooding and high bed load transport through erosion of the upstream debris flow deposits) cannot be excluded.





The multi-methods approach for debris flow hazard scenario building and runout modelling in pro- and periglacial catchments presented here is easily applicable to other catchments, also by practitioners. It is among the first to propose a methodology for the definition of precipitation dependent triggering scenarios and allows sound estimates of potential debris flow volumes based on both empirical methods and detailed field investigations, limiting the uncertainties inherent to both approaches. As a result, PPDF scenarios including potential triggering events and assigned return periods can be defined with a high level of detail, providing valuable information for debris flow hazard assessment and risk management.

*Code and data availability.* The sources of the data and methods used in this study are described in chapters 3 and 4 of the article. The data should be requested through the mentioned institutions or downloaded from the provided websites.

**Appendix A: Geomorphological characteristic sections of the Dar torrent**

The uppermost part of the Dar catchment is characterized by glacier- and permafrost-related processes, with the rapidly shrinking very small Glacier du Sex Rouge (Fischer et al., 2016; Huss and Fischer, 2016; Fischer, 2018; GLAMOS, 2021) and the likely extensive permafrost occurrence in the northwestern flank of Oldehore/Becca d'Audon (3123 m a.s.l.) and steep slopes east of Sex Rouge (2972 m a.s.l.) (FOEN, 2005, Fig. 2b, Fig. A1a).

Between the first and second rock step (Fig. 1a,c, Fig. A1), the Dar crosses the steep (~45%) forefield of Glacier du Sex Rouge situated in the cirque of Le Dar Dessus (chapter 2.2, Fig. 2, Fig. A1b). Several gullies (secondary channels) carrying water in case of heavy rainfall or snow and ice melt cut through the extensive pro- and periglacial sediment deposits, and converge with the main channel towards the second steep rock step.

Just below the second steep rock step is a fan-like landform of intermediate sediment deposits (Fig. A1c) representing maximum runout for low magnitude, high frequency debris flow events starting in the cirque of Le Dar Dessus. After the confluence of a steep gully showing debris flow activity in 2005 (Fig. 3), the Dar becomes more channelized and crosses a still relatively steep (~30%) alpine meadow (Fig. 1c, Fig. A1d). In this section of the torrent, further below where crossing the third steep rock steep (Cascade du Dar) and until the sharp left turn and abrupt change in slope, the narrow and incised channel bed more or less consists of bare bedrock (mostly bright limestones) (Fig. A1d,e).

Downstream from the sharp left turn, the Dar follows a much more gently inclined (ca. 8%) valley floor westwards until the village of Les Diablerets (Fig. 1c). At Creux du Pillon, the torrent has deeply incised into the debris flow deposits of the summer 2005 event. Most of the area of debris flow deposits – especially where the debris flow broke out of the channel in 2005 (Fig. 3, Fig. 7) – is now covered by pioneer species, willow and birch. The channel bed consists of easily mobilizable loose sediments (lots of fine-grained material, gravel and blocks) (Fig. A1f).

Between Creux du Pillon and Les Diablerets, the Dar mostly runs in in a forested, typical v-shaped fluvial valley. A number of shallow embankment slides are characteristic of these sections of the torrent (Fig. A1g). Unlike for the Creux du Pillon area, the channel bed is mainly composed of gravel and blocks. Fine-grained sediments are continuously transported downvalley





during high water. About 850 m before its confluence with La Grande Eau at Les Diablerets, a tributary torrent (Ruisseau des

Roseyres) converges with the Dar (Fig. A1). The former flows through an area of active landslides (Jaboyedoff et al., 2009).

In case of heavy precipitation events, landslides and embankment erosion can occur in this zone. Thus, important volumes of

loose material can be transported towards the confluence with La Grande Eau (B+C Ingénieurs SA, 2013).

   After the confluence of Ruisseau des Roseyres, the channel bed of the Dar widens, and both slope and embankment processes

become less important (Fig. A1h). A ca. 250 m long and 50 m wide bed load retention area was created around 2017 just 200

m upstream of the confluence of the Dar with La Grande Eau (Fig. A1i, Fig. 10). The retention area is dimensioned for an

event of a magnitude similar to the summer 2005 event (B+C Ingénieurs SA, 2013).

## Appendix B: Assessment of glacier- and permafrost-related hazards above the first steep rock step

### B1    Formation of a new lake at Glacier du Sex Rouge

The very small Glacier du Sex Rouge is located in the uppermost section of the Dar catchment, just above the first steep rock

step (Fig. 1a,c). It has shown rapid and accelerating mass loss and shrinkage during the past decades (Fischer et al., 2015, 2016;

Fischer, 2018; GLAMOS, 2021) and is expected to have completely disappeared by 2035 (Huss and Fischer, 2016; Fischer

and Keiler, 2019). Since 2016, a new proglacial/ice-marginal lake has started to form at the location of an incision into the first

rock step, where the glacial meltwater drains towards the cirque of Le Dar Dessus (upper photograph in Fig. B1d). The lake is

currently growing in size every summer, enhancing accelerated glacier shrinkage.

   The future area, depth and volume of the lake were estimated using a combination of ice thickness measurements of 2010

(courtesy of Matthias Huss, cf. Gärtner-Roer et al., 2014) and 2012 (Fischer, 2018) with glacier outlines from 2010 (Fischer

et al., 2014) and a high-resolution (2 m) 2010 swissALTI[3D] DEM from swisstopo (Fig. B1a,b). The ice thickness data were

collected in the winters of 2010 and 2012 using a portable Måla ground penetrating radar (GPR) ProEx System and a 50 or

100 MHz antenna as well as a conventional GPS to capture the measurement positions. The raw GPR data were analysed in

Reflexw 2D (Sandmeier Scientific Software) applying a series of standard processing steps (Sold et al., 2013). With the help

of measured annual mass balance data (GLAMOS, 2021), the 2012 ice thickness profiles were time corrected to represent

2010 values. The linear ice thickness profiles and the digital glacier outline (where ice thickness equals zero, except for Col de

Tsanfleuron (lower photograph in Fig. B1d)) were then used to calculate the spatially distributed 2010 ice thickness of the entire

930    glacier (Fig. B1a). The uncertainty in the latter (as a result of uncertainty in the GPR measurements, GPR data processing, and

spatial interpolation) is less than $\pm 20\%$.

   Subsequently, the 2010 ice thickness data were subtracted from the 2010 swissALTI[3D] DEM to derive the glacier bed

topography and thus calculate the future area (ca. 27'500 m$^2$), volume (ca. 250'000 m$^3$) and depth (mean: 9.2 m; max.: 14.5

m) of the new lake (Fig. B1b). Based on the mapped lithological units surrounding Glacier du Sex Rouge and their dip angles

935    (geological map 1:25'000 "Les Diablerets", Badoux and Gabus, 1991), a geological profile was drawn across the future lake

(red dashed line A–B in Fig. B1b) to assess the permeability of the substrate in the topographic depression where the lake is

expected to form. Due to the weak permeability of the underlying siliceous and sandy limestones as well as marly shales and





clayey limestones (Fig. B1c), the water level (at ca. 2742 m a.s.l.) and geometry (Fig. B1b) of the future lake are expected to remain stable as soon as its formation will be completed. Independent of our work, Neven et al. (2021) carried out GPR surveys on Glacier du Sex Rouge in 2018. Their measured ice thickness data and basal topography estimation confirm the existence of a shallow glacial overdeepening or topographic depression at the very same location where we expect the lake to grow.

## B2   Assessment of future rock fall impact disposition into the new lake

The anticipated complete disappearance of Glacier du Sex Rouge by 2035 (Huss and Fischer, 2016) and formation of a relatively small and shallow new lake just above the first steep rock step (chapter B1, Fig. B1) raises questions related to potential new hazards (here especially glacier lake outburst floods (GLOFs)) and process interactions for the downstream area of the Dar catchment (e.g., Haeberli et al., 2016; Mani et al., 2022). The new lake will be impounded by bedrock and will have little to no freeboard (Fig. B1b,d). Therefore, the possible outburst mechanisms that could potentially cause flooding further downvalley would be overtopping or overflow (Haeberli, 1983; Schneider et al., 2014), the former being more hazardous and thus relevant to assess here.

An outburst of the lake by overtopping of the rock dam could only be caused by gravitational processes from the northwestern flank of Oldehore/Becca d'Audon (Fig. 1a,c, lower photograph in Fig. B1d) reaching the lake and causing an impact wave. Hence, the rock fall impact susceptibility into the new lake was assessed following Schaub (2015) and applying a GIS-based multi-criterion evaluation of different disposition parameters (Fig. B2). Decisive factors are the slope angle and the potential runout distance of such processes. Within the watershed of the future lake, the two basic requirements were assessed using a high-resolution (2 m) 2016 swissALTI$^{3D}$ DEM from swisstopo. They were considered fulfilled for slopes of potential rock detachment zones steeper than 30° with an overall trajectory slope >25° between the detachment zone and the lake ("topographic potential" in Fig. B2).

Three additional parameters (potential permafrost distribution, lithology, and deglaciated area since 1850) were analysed for the catchment of the future lake. These parameters are considered non-decisive but with a certain influence on the susceptibility of rock falls as they influence slope stability. Modelled data on future potential permafrost distribution were provided by Deluigi et al. (2017). Areas with a probability of >50% for the presence of permafrost in 2035 were assigned an additional fulfilled parameter for the rock fall disposition (Fig. B2), assuming that temperature-induced permafrost degradation will decrease slope stability. Indeed, over the last few years, an increase in rock fall activity and hence supraglacial debris cover on Glacier du Sex Rouge has been observed, possibly indicating increasing instability of the rock face due to permafrost warming. For the different lithologies present in the area (geological map 1:25'000 "Les Diablerets" from Badoux and Gabus (1991) and Fig. B1c), only the siliceous limestones of the Helvetic Kieselkalk Formation were assumed to contribute to the rock fall disposition (Fig. B2), since the other lithologies (marl-limestone alternation, marly shales and clayey limestones) generally produce fine-grained debris. The effects of slope destabilization through deglaciation were accounted for areas between the 1850 and 2010 glacier outlines (SGI1850 and SGI2010, Maisch et al., 2000; Paul, 2004; Fischer et al., 2014) (Fig. B2).





In summary, calculated following Schaub (2015), it can be said that the highest disposition for rock fall processes reaching
the future lake can be assigned to the very steep (>40°) parts of the northwestern flank of Oldehore/Becca d'Audon composed
of siliceous limestones (Fig. B2). In contrast to Schaub (2015), lithology is a rather decisive parameter here.

A rock fall event in which a relevant rock volume would break out of the rock face as a single and cohesive package is only
expected for the siliceous limestones of the Helvetic Kieselkalk Formation. In addition, the approximately 25° dip towards
southeast of the various lithologies forming the Oldehore/Becca d'Audon (Fig. B1b,c) significantly reduces the potential for
larger rock fall events to the northwest and hence into the future lake, as dip slopes are known to bear a higher potential for rock
slides compared to scarp slopes. Indeed, two rock fall events out of the Helvetic Kieselkalk Formation have occurred on the
dip slope side of Oldehore/Becca d'Audon during the past years (Fig. B3c,d). It is assumed that in the worst case, triggered by
increased pore water pressure or permafrost warming, one of the pillar- or spike-like bulges in the scarp slope of the siliceous
limestone formation of the northwestern flank of Oldehore/Becca d'Audon could break off as a compact rock package and
fall northwestwards into the lake (Fig. B3a,b). In such a case, the maximum potential rock fall volume for a single event is
estimated to range between few $10^2$ to few $10^3$ m$^3$.

### B3    Assessment of GLOF potential through rock fall into the new lake

To assess the downstream effects of future potential GLOFs triggered by rock fall generated impulse waves, an Excel-based
tool provided by Evers et al. (2019) for manual computation of impulse wave generation and its effects on dams was used. To
compute the generation and propagation of a three-dimensional impulse wave, the following parameters were calculated based
on (i) the previously defined future potential rock fall events, (ii) the future ice-free topography and lake extent (Fig. B1b), and
(iii) equations in Evers et al. (2019). For the latter, parameters determining the defined rock fall event (bulk slide thickness,
width, volume, density and porosity; slide impact velocity, impact angle into the lake; still water depth), parameters defining
the wave propagation (radial distance, wave propagation angle), and parameters defining the shore/dam situation (run-up angle,
freeboard, dam crest width) had to be defined. According to our calculations, a rock fall of 1000 m$^3$ reaching the lake would
cause an outburst flood of 625 m$^3$ that would overflow the incision into the first steep rock step of the Dar catchment at the
lake outflow (upper photograph in Fig. B1d) for about 13 seconds and by maximal 2 m (Table B1). Especially compared to the
clear water discharges characterizing the precipitation dependent triggering events defined for our PPDF scenarios (Table 4),
such a GLOF was judged of too small magnitude to be able to trigger a major debris flow event in the cirque of Le Dar Dessus.

### Appendix C:  Calculating clear water discharge per triggering scenario with ZEMOKOST

In order to run ZEMOKOST, a number of catchment characteristics had to be defined first (Table C1). The catchment area
for which clear water discharge was calculated (red contours in Fig. C1) was computed using a high-resolution (2 m) 2016
swissALTI$^{3D}$ DEM from swisstopo and amounts to 2.1 km$^2$. The mean length and mean slope of channelless surface runoff
from the watershed boundaries to the main channel were calculated from the average values of 10 segments shown in Fig. C1a
and result in 937 m and 0.61 (27.5°). Baseflow was approximated from data on modelled mean monthly runoff for the Dar



**Table B1.** Rock fall triggered outburst scenarios for the future lake at today's location of Glacier du Sex Rouge calculated with an Excel-based tool for manual computation of impulse wave generation provided by Evers et al. (2019).

| Rock fall volume | Lake volume | Lake outburst | | | Remarks |
|---|---|---|---|---|---|
| | | Overflow time | Max. overflow depth | Outburst volume | |
| 1'000 m³ | 250'000 m³ | ca. 13 seconds | ca. 2 m | ca. 625 m³ | Assumed largest rock fall volume to be expected with a certain probability that could break off as a compact rock fall in a single event |
| A few 1'000 m³ | 250'000 m³ | A few $10^1$ seconds | A few meters | A few 100 to few 1'000 m³ | The higher the volume, the more unlikely |

(Fig. 1d, chapter 2.1, FOEN, 2000). For thunderstorm and long-lasting rainfall scenarios, the average value of 0.7 m³/s during April to September was adopted, and for ROS scenarios the average value from May to July (0.9 m³/s) was applied.

To determine the percentage area of the catchment belonging to predefined runoff and roughness classes, (i) different types of landcover were mapped using high-resolution (0.1 m) 2020 SWISSIMAGE Level 3 orthophotos from swisstopo, and (ii) runoff and Manning's roughness coefficients were assigned to each landcover type (Fig. C1b). For all ROS scenarios, runoff coefficients were set to 1.0 as complete saturation of the substratum (sediments) due to snow melt was assumed. Based on field observations in the cirque of Le Dar Dessus (rather coarse material with an important fine-grained proportion), intermediate runoff was set to 1.0 ("very permeable"). The relative amount of intermediate runoff was calculated using the percentage area of the different landcover types and their runoff coefficients (Fig. C1b). For all ROS scenarios, the latter was set to 0% (no intermediate runoff), again assuming complete saturation of the substratum (sediments) due to snow melt.

The final input parameters characterizing the catchment necessary for the calculations with ZEMOKOST relate to the topology of the main channel (Table C1, Fig. C1). Grain size distribution was estimated in the field and based on high-resolution (0.1 m) 2020 SWISSIMAGE Level 3 orthophotos from swisstopo. The value for "d90" was set to 0.5 m (i.e. 90% of the individual particles were estimated to be smaller than 0.5 m in length). After determining the catchment characteristics and input parameters (Table C1), data on rainfall respectively snow cover runoff intensities and duration (Table 1 and Table 3 in chapter 3.1) were imported into ZEMOKOST. Subsequently, clear water discharge was computed for all thunderstorm, long-lasting rainfall and ROS scenarios.




**Table C1.** Catchment characteristics of the cirque of Le Dar Dessus and necessary input parameters in ZEMOKOST for the calculation of clear water discharge and hydrographs per meteorological triggering scenario.

| | Surface characteristics | | | | Belonging to runoff classes defined in ZEMOKOST (% of catchment area) | | | | | | |
|---|---|---|---|---|---|---|---|---|---|---|---|
| | Area (km$^2$) | Mean length (m) | Mean slope (1 = 45°) | Baseflow (m$^3$/s) | 0 | 1 | 2 | 3 | 4 | 5 | 6 |
| Thunderstorm and long-lasting rainfall scenarios | 2.10 | 937 | 0.61 | 0.7 | - | - | 50 | 13 | 37 | - | - |
| ROS scenarios | 2.10 | 937 | 0.61 | 0.9 | - | - | - | - | - | - | 100 |

| | Belonging to roughness classes defined in ZEMOKOST (% of catchment area) | | | | | | Intermediate runoff | | Topology of the main channel | | |
|---|---|---|---|---|---|---|---|---|---|---|---|
| | 1 | 2 | 3 | 4 | 5 | 6 | 1 to 7 | Proportion (%) | Length (m) | Slope (1 = 45°) | Grain size (d90 (m)) |
| Thunderstorm and long-lasting rainfall scenarios | - | - | 50 | - | 50 | - | 1.0 | 57.0 | 1334 | 0.48 | 0.50 |
| ROS scenarios | - | - | 50 | - | 50 | - | 1.0 | 0.0 | 1334 | 0.48 | 0.50 |

## Appendix D: Bed load volume estimations for PPDFs starting in the cirque of Le Dar Dessus

To estimate bed load volumes for PPDFs starting in the cirque of Le Dar Dessus according to the approach for small and steep torrent catchments proposed by Spreafico et al. (1996), the decision tree shown in Fig. D1 for qualitative estimation of the expected bed load volume for an event with a return period of 100 years was used first. For the present study, a 100-year event equals a medium event size (DF$_{medium}$, Table 7). For the cirque of Le Dar Dessus, the formation of debris flows is clearly possible, as the summer 2005 event has shown (chapter 2.3). Indeed, for the initiation area, the average slope of the main

channel is >20% and channel sections with slopes well below 20% do not exist (Fig. 1c). The cirque of Le Dar Dessus is hence clearly prone to debris flows. In the immediate surroundings of the (main) channel, large sediment deposits do exist, whereas relatively long channel sections made up of bedrock are absent (Fig. 2a, Fig. C1b). Due to the incised channels (Fig. D2d,e) and steep slopes, it is argued here that substantial intermediate deposition of sediments in the channel does not take place during





**Table D1.** Specific bed load volumes for areas dominated by limestones depending on the size of the catchment area (after Spreafico et al., 1996).

| Size of the catchment area | Specific bed load volume (m³/km²) | | | | |
|---|---|---|---|---|---|
| | very small | small | medium | large | very large |
| 1 – 10 km² | 800 – 200 | 1'000 – 500 | 5'000 – 1'000 | 10'000 – 2'000 | 30'000 – 3'000 |

**Table D2.** Determination of multiplication factors to calculate bed load volumes for small ($BL_{small}$) and large ($BL_{large}$) debris flow events starting in the cirque of Le Dar Dessus from the previously determined bed load volume for medium-sized ($BL_{medium}$) events. Calculated according to Spreafico et al. (1996) (cf. Hunziker et al., 2014).

| Individual factors | | $BL_{small}$ | $BL_{large}$ |
|---|---|---|---|
| Dominant transport process in the catchment | here: debris flow | 0.3 – 0.5 (here: 0.5) | 1.8 – 2.0 (here: 2.0) |
| Size of the catchment area | 0.5 – 5 km² | 0.4 – 0.6 (here: 0.5) | 1.2 – 1.7 (here: 1.5) |
| Relatively long channel sections of bedrock exist | here: no | 0.3 – 0.5 (here: 0.5) | 1.5 – 2.0 (here: 2.0) |
| Large sediment deposits in immediate surroundigs of the channel exist | here: yes | 0.3 – 0.5 (here: 0.5) | 1.5 – 2.0 (here: 2.0) |
| Substantial intermediate material deposition during event | here: no | 0.3 – 0.5 (here: 0.5) | 1.5 – 2.0 (here: 2.0) |
| Threshold processes* | here: potentially possible | 1.0 | 2.0 – 5.0 (here: 3.0) |
| Calculation of final multiplication factor $mf$ from the average of all individual factors | here: $mf_{small} = 0.583$, $mf_{large}$ = 2.083 | $BL_{small}$ = 0.583 * 54'000 ≈ 31'000 m³ | $BL_{large}$ = 2.083 * 54'000 ≈112'500 m³ |

*In rare cases, threshold processes can occur which can increase runoff and bed load volume by an order of magnitude. Threshold processes are mainly triggered during large or very large events.

a medium-sized debris flow event starting in the cirque of Le Dar Dessus. Ultimately, these considerations resulted in a "very

large" expected specific bed load volume for a medium-sized event (brown path in Fig. D1).

    In areas dominated by limestones (as the cirque of Le Dar Dessus, Fig. 1b), specific bed load volumes decrease with increasing size of the catchment area (Spreafico et al., 1996, Table D1). As the expected specific bed load volume for a medium-sized debris flow is very large for the cirque of Le Dar Dessus (Fig. D1), the resulting value is 27'000 m³/km² (Table 5) (derived from linear interpolation of the "very large" specific bed load volumes in Table D1 to a size of the catchment area of roughly 2 km²,

Table C1). Applying the approach proposed by Spreafico et al. (1996), the resulting bed load volume for a medium-sized debris




flow event is hence 2 x 27'000 m$^3$ = 54'000 m$^3$ for the cirque of Le Dar Dessus. Rounded to the next higher five thousand, BL$_{medium}$ equals to 55'000 m$^3$ (Table 5).

Finally, bed load volumes for small (BL$_{small}$) and large (BL$_{large}$) debris flows starting in the cirque of Le Dar Dessus were calculated from the multiplication of the previously derived bed load volume for a medium-sized event (BL$_{medium}$) by the average values of different factors known to impact bed load volume (Table D2). The values for these individual factors were assessed for the cirque of Le Dar Dessus based on our understanding of the relevant characteristics, geomorphological processes and functioning of the catchment, obtained through a combination of available geodata (in particular high-resolution aerial orthophotos and DEMs) and field surveys. Rounded to the next higher five thousand, BL$_{small}$ equals to 35'000 m$^3$, and BL$_{large}$ to 115'000 m$^3$ (Table D2, Table 5).

For plausibility check and to complement bed load volume estimates for different event sizes derived according to Spreafico et al. (1996) and bed load potentials calculated based on empirical formulas for the cirque of Le Dar Dessus (Table 5, Fig. D1 and Tables D1-D2), SEDEX$^©$ (SEDiments and EXperts, Frick et al., 2011) was applied. SEDEX$^©$ is a standardized, practice-oriented assessment tool for field-based estimations of event-specific debris flow bed load volume supply at the fan apex of small (<10 km$^2$) alpine torrent catchments. The torrent system has to be divided into (geomorphologically) more or less homogenous channel sections, for which channel, embankment and slope processes with respect to erosion, transport and deposition of loose material are assessed. Subsequently, for different meteorological triggering scenarios, the volume of delivered solids is calculated for individual channel sections and the whole torrent system (Fig. D2a,b). A user manual and several checklists (Frick et al., 2011) systematically guide practitioners through all the necessary working steps in the field and with the provided software.

Here, the amount of bed load volume that could potentially be released in the cirque of Le Dar Dessus during a large event was assessed using SEDEX$^©$. Therefore, the torrent system of the cirque, consisting of a main and several secondary channels, was divided into a western (section 1) and eastern (section A1) part (Fig. D2c). Erosion in unlimited material is expected to be the dominant process for both slope, embankment and channel parts of the cirque of Le Dar Dessus during a large debris flow event. The corresponding total amount of mobilised bed load volume leaving the cirque of Le Dar Dessus (i.e. bed load volumes for sections 1 and A1 together) resulted in 121'572 m$^3$ (Fig. D2d,e). Hence, rounded to the next higher five thousand, BL$_{large}$ estimated applying SEDEX$^©$ equals to 125'000 m$^3$ (Table 5).

## Appendix E: Quality assessment of calibrating RAMMS-DF to the summer 2005 PPDF event

For quality assessment of the calibration of RAMMS-DF to the summer 2005 debris flow event and, hence, to the Dar catchment, the area of the simulated debris flow deposits after final calibration (DF$_{c\_2005}$, Tables 13-14) was superimposed to the area of the mapped debris flow extents at Creux du Pillon (Fig. 7). The comparison of these areas resulted in four different classes (true positive, false positive, false negative, true negative, Fig. E1). About half of the total area (50.2%) could be assigned to the true positive class (both mapped and simulated debris flow deposits), 41% of the total area was simulated as debris flow



deposits outside the actual perimeter of the mapped 2005 event deposits (false positive), and 8.8% of the total area was assigned to the false negative class (mapped but not simulated debris flow deposits).

Three different model performance indices for hazard assessment were calculated according to Beguería (2006). The total proportion of correctly simulated areas (*model efficiency*) resulted in 0.75 (75%). The proportion of correctly predicted positive cases (TP) respectively the predictive power of the model (*model sensitivity*) was good (0.85 respectively 85%). The proportion of correctly predicted negative cases (TN) characterizes the model as pessimistic or optimistic (*model accuracy*). A model that neither performs too pessimistically nor too omptimistically shows accuracy indices close to 0.75. Here, with an accuracy index

of 0.71, the calibration of RAMMS-DF to the summer 2005 debris flow event in the Dar catchment neither overestimated nor underestimated the areas of mapped debris flow deposits. Therefore, it can be said that, after model calibration, simulations of the PPDF scenarios for the Dar are neither too optimistic nor too pessimistic (accuracy index close to 0.75) (Fig. E1).

*Author contributions.* MF initiated and led the research project of which this study is part of. With the help of CG and MF, RA calibrated RAMMS-DF to the 2005 debris flow event in the Dar, developed the methodological approach for scenario building for PPDFs and produced

modelling results for the debris flow scenarios. CG provided assistance for technical issues with RAMMS-DF and debris flow modelling in general. AR carried out analyses of bed load volumes with SEDEX© and the geomorphological functioning of the Dar catchment. PS provided important information and data on the study area in general and the 2005 debris flow event in particular. MKe and MZ helped with different inputs and expertise throughout the study. MF drafted the manuscript together with MKu. MF, MKu and RA created all figures. MF prepared all appendices. All authors contributed to the final version of the manuscript.

*Competing interests.* At least one of the (co-)authors is a member of the editorial board of Natural Hazards and Earth System Sciences.

*Acknowledgements.* This study was funded by the University of Bern, Switzerland. The license and full version of RAMMS-DF was provided by the Swiss Federal Institute for Forest, Snow and Landscape Research (WSL, Birmensdorf, Switzerland). Many thanks to Luc Braillard (Department of Geosciences, University of Fribourg, Switzerland), for geological advice and expertise, Alexander Groos (Institute of Geography, Friedrich-Alexander-University Erlangen-Nuremberg (FAU), Erlangen, Germany), for the acquisition of UAV data for

the cirque of Le Dar Dessus and assistance in the field, Frédéric Guex (B+C Ingénieurs SA, Montreux, Switzerland), for providing us the photo documentation and practitioners' reports of the 2005 event, Julien Viquerat (Office de l'information sur le territoire, Canton of Vaud, Lausanne, Switzerland), for supplying us with the high-resolution ALS-DEMs of 2001/02/05 for the Dar catchment, and Glacier 3000 AG (Les Diablerets, Switzerland), for free cablecar transportations for fieldwork in the framework of this research project.



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





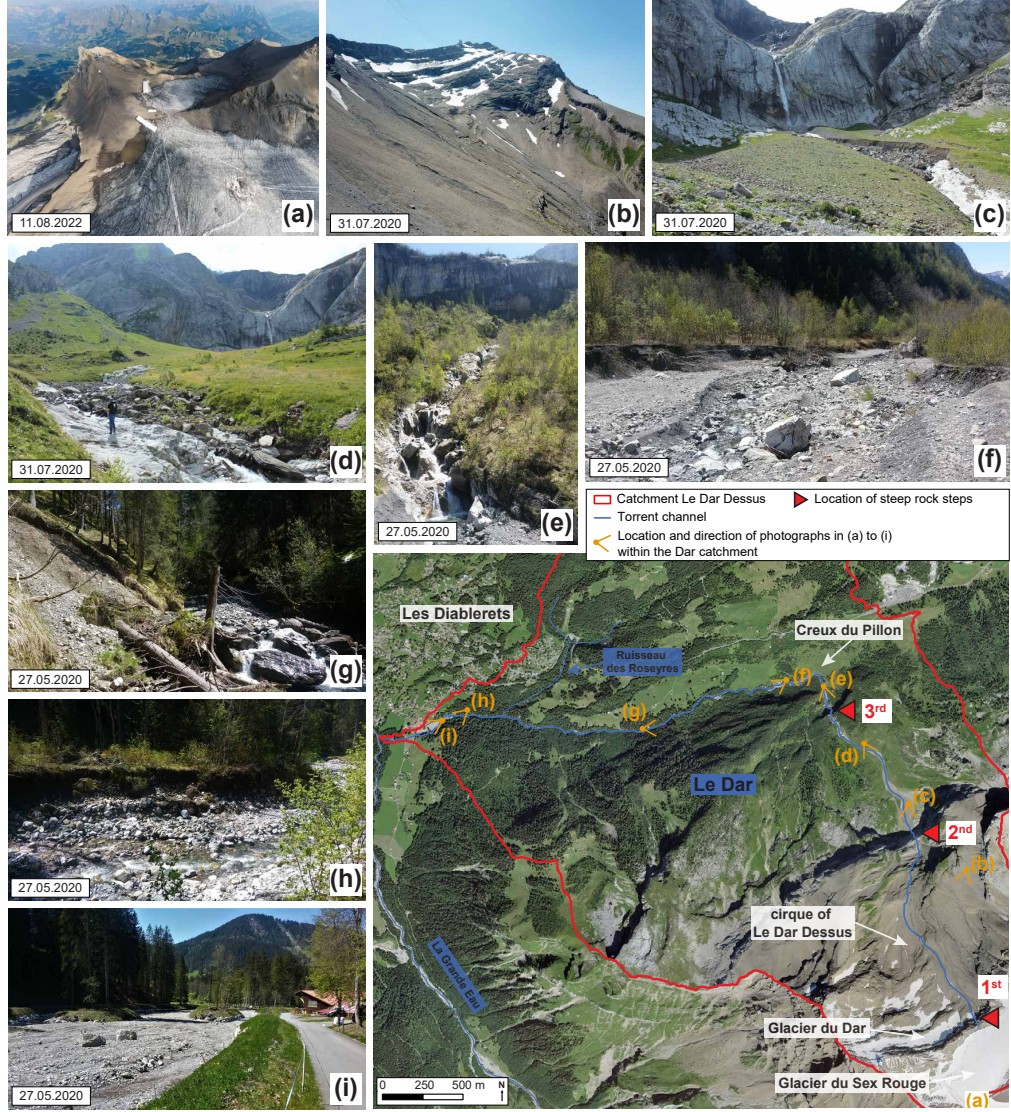

**Figure A1.** The Dar catchment. Recent situation of geomorphological characteristic torrent sections from source to mouth. (a) Uppermost parts of the catchment with northwestern flank of Oldehore/Becca d'Audon, Glacier du Sex Rouge, and Sex Rouge summit (from right to left). Drone image taken from above Glacier de Tsanfleuron in August 2022 (courtesy of Glacier 3000 AG). (b) The steep cirque of Le Dar Dessus with the first rock step and Sex Rouge summit in the background. (c) Incised fan-like intermediate sediment deposits just below the second steep rock step. (d) The Dar crossing an alpine meadow in open terrain. The second rock step is visible in the background. (e) The third steep rock step (Cascade du Dar) with the Dar torrent incised into bright limestones. (f) Lower part of incised debris flow deposits of the summer 2005 event at Creux du Pillon. (g) Shallow embankment slide on the orographic right side of the Dar torrent at 1313 m a.s.l. (h) Widening of the torrent channel after the confluence of Ruisseau des Roseyres. (i) Bed load retention area at Les Diablerets, few hundreds of meters upstream of the confluence of the Dar with La Grande Eau. The location and direction of photographs in (b) to (i) taken by A. Rüeger on May 27 and July 31, 2020 is indicated with orange symbols on the overview map that is underlain with a 2020 SWISSIMAGE orthophoto from swisstopo.



**Figure B1.** Formation of a new lake at Glacier du Sex Rouge. (a) 2010 ice thickness distribution derived from GPR surveys in 2010 and 2012 (Gärtner-Roer et al., 2014; Fischer, 2018). (b) Glacier bed topography calculated from the subtraction of (a) from a 2010 swissALTI³ᴰ DEM from swisstopo. The expected future lake outline (at ca. 2742 m a.s.l.) and its surface outflow towards the cirque of Le Dar Dessus are marked in blue. Underlain is a 2016 SWISSIMAGE orthophoto from swisstopo. (c) Geological profile along the red dashed line A–B in (b) derived from the geological map 1:25'000 "Les Diablerets" (Badoux and Gabus, 1991). (d) Above: incision into the first steep rock step of the Dar catchment with surface outflow of the forming proglacial/ice-marginal lake (photograph by M. Fischer, 20.08.2020). Below: northwestern flank of Oldehore/Becca d'Audon, Glacier du Sex Rouge, forming lake and first steep rock step (photograph by M. Fischer, 01.08.2022).



**Figure B2.** Assessment of the future rock fall impact disposition from the northwestern flank of Oldehore/Becca d'Audon into the new lake applying a GIS-based multi-criterion evaluation of different disposition parameters following Schaub (2015). Photograph by M. Fischer, 11.09.2014.





**Figure B3.** Rock fall potential and past events from the siliceous limestones (Helvetic Kieselkalk Formation) at Oldehore/Becca d'Audon. (a) Pillar- or spike-like bulges in the scarp slope side (northwestern flank) bearing a certain potential for rock falls into the future lake, triggered by enhanced pore water pressure or permafrost warming. Photograph by M. Fischer, 22.09.2014. (b) Cross-sectional profile B–A' (Fig. B1b) for a rock fall trajectory reaching the future lake and causing an impact wave that could lead to a small lake outburst by overtopping the incision at the first steep rock step of the Dar catchment. (c) Detachment zone (orange rectangle) and deposits of ca. 7'000 m$^2$ of a rock fall event around 1992 (swisstopo, 2022) on the dip slope side (southern flank) of Oldehore/Becca d'Audon, close to Col de Tsanfleuron. Photograph by M. Fischer, 15.09.2019. (d) Deposits of ca. 55'000 m$^2$ of a rock fall event in 2019 (swisstopo, 2022) on the dip slope side of Oldehore/Becca d'Audon, ca. 100 m east of the 2022 terminus position of Glacier de Tsanfleuron. Photograph by M. Fischer, 15.09.2019.



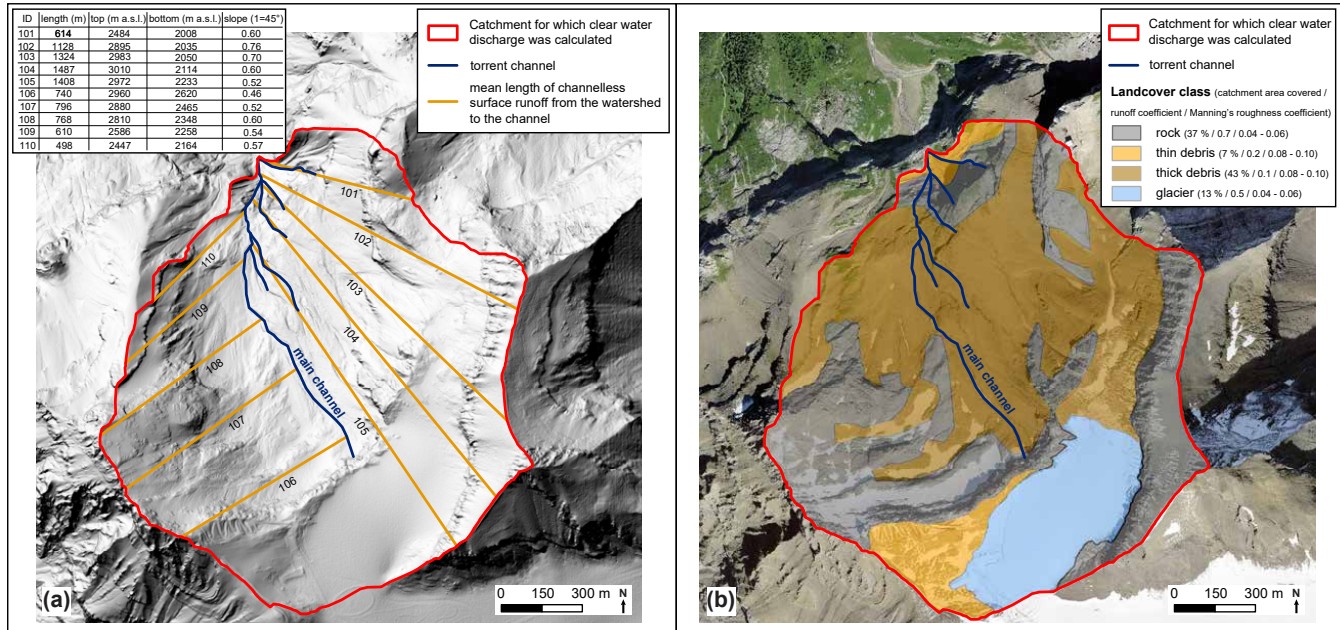

**Figure C1.** Hydrological catchment of the cirque of Le Dar Dessus for which clear water discharge and hydrographs were calculated for each meteorological triggering scenario using version 2.0 of ZEMOKOST (Kohl and Stepanek, 2005; Kohl, 2010). (a) Catchment area and channelless surface runoff from the watershed boundaries to the main channel. Underlain is a hillshade of the 2016 swissALTI$^{3D}$ DEM (2 m resolution) from swisstopo. (b) Defined landcover classes within the catchment, including assigned runoff and roughness coefficients. Underlain is a 2020 SWISSIMAGE orthophoto (0.1 m resolution) from swisstopo.



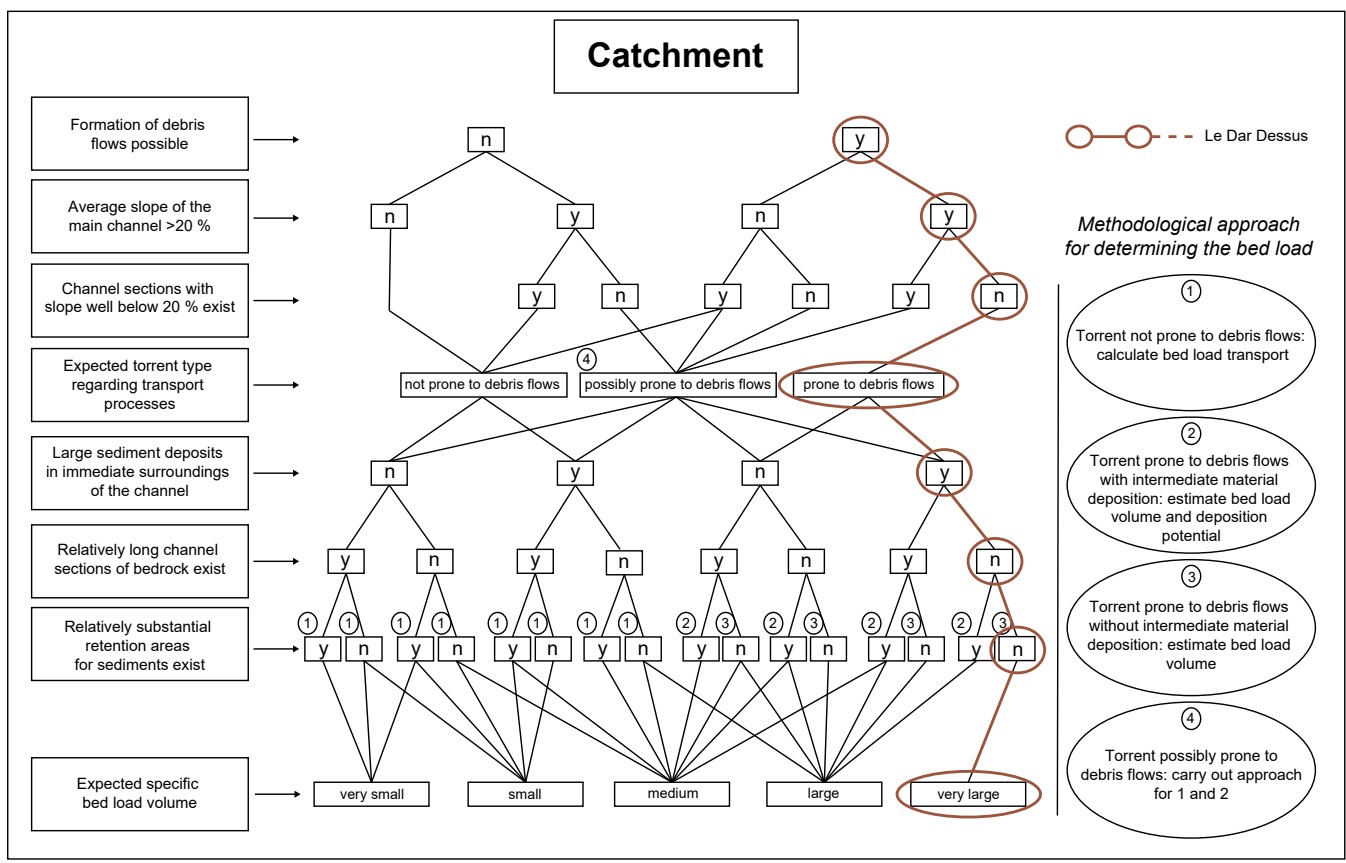

**Figure D1.** Decision tree for qualitative estimation of bed load volumes for medium-sized events ($BL_{medium}$) in small and steep torrent catchments (adopted and supplemented from Spreafico et al., 1996). The chosen path and expected (qualitative) specific bed load volume for a medium-sized event starting in the cirque of Le Dar Dessus is highlighted in brown.





**Figure D2.** Application of SEDEX© (SEDiments and EXperts, Frick et al., 2011) to the cirque of Le Dar Dessus to calculate event-specific bed load volumes. (a) Different sediment sources and processes to be assessed for the slope, embankment and channel parts of individual torrent sections and (b) schematic diagram for calculation of downstream sediment delivery from slope, embankment and channel modules for a sequence of torrent sections. (c) Catchment of the cirque of Le Dar Dessus with channel sections for which SEDEX© was applied. The location and direction of photographs taken of both sections (see (d) and (e)) is indicated with orange symbols. Underlain is a hillshade of the 2016 swissALTI³D DEM (2 m resolution) from swisstopo. (d) Situation of section 1 in the lower part of the cirque on 31.07.2020 (photograph by A. Rüeger) and resulting bed load volume from section 1 for a large debris flow event. (e) Situation of section A1 in the lower part of the cirque on 31.07.2020 (photograph by A. Rüeger) and resulting bed load volume from section A1 for a large debris flow event.



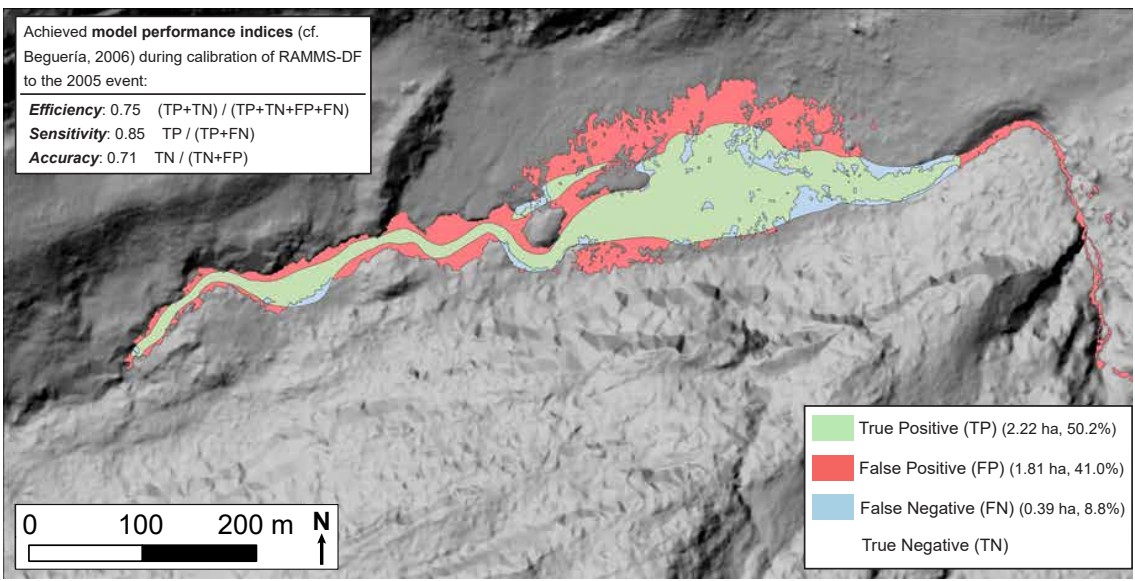

**Figure E1.** Quality assessment of calibrating RAMMS-DF to the Dar based on the mapped extents of the summer 2005 debris flow event (Fig. 7). Correctly simulated debris flow extents (true positive, TP) are shown in green, areas erroneously simulated as debris flow deposits (false positive, FP) in red, and areas erroneously simulated as not belonging to the extents of debris flow deposits (false negative, FN) in blue. In addition, resulting values of different model performance indices (Beguería, 2006) achieved by the final calibration of RAMMS-DF to the Dar catchment are shown. Underlain is a hillshade of the 2001/02 ALS-derived DEM (1 m resolution) from the Office de l'information sur le Territoire (2002), Canton of Vaud.