# Peer review of "Scenario building and runout modelling for debris flow hazards in pro-/periglacial catchments with scarce past event data: application of a multi-methods approach for the Dar catchment (western Swiss Alps)"

_EGUsphere, 2023_

## Author Comment (AC1)

**Reply to RC1 (Stefan Hergarten) "Comment on egusphere-2023-1190" by Mauro Fischer et al.**

In *italic*: Stefan Hergarten's comments

In blue normal font: our replies

*This paper provides a hazard assessment with regard to debris flows for a small catchment in the Alps. The results are finally obtained from simulations with the debris flow version of the model RAMMS. Focus is on simulating different scenarios, defined by different amounts of precipitation and available sediment.*

*The manuscript is very long. I often had the feeling that it reads like a thesis rather than like a scientific paper. I guess that it is indeed a shortened version of the M.Sc.- thesis of the third author (Reto Aeschbacher). I am quite sure that the thesis was excellent and that the supervisors were delighted about it. But on the other hand, it is not concise and not to the point. Each step is expanded in great detail, which is fine in a thesis, but not in a scientific paper.*

Thank you very much for your comment. We generally agree with the reviewer here and will shorten the manuscript and develop a clearer structure to highlight the crucial parts of the research. The first submitted version of our manuscript has somewhat "guidelines" character and is therefore very long. Our rationale here was to provide researchers and practitioners a workflow with detailed description of important procedures one has to work through for debris flow scenario building and numerical runout modelling, especially for pro-/periglacial catchments with inexistent or scarce past event data. To our knowledge, for pro-/periglacial debris flows triggered by precipitation dependent events, such comprehensible/understandable procedures have so far been lacking (thus the added value of our research). We kept a lot of details in the first version of our manuscript because, from our experience, important steps are often missing or unclear in existing literature on debris flow scenario building and numerical runout modelling in general, and do, to our knowledge, not exist yet in a comprehensive manner for pro-/periglacial catchments. Nevertheless, we will shorten the manuscript wherever possible and provide references for individual working steps. Where we cannot provide references to some of the details/steps in the proposed workflow, we will leave relevant additions/descriptions in the text (but as brief as possible).

*As a main point, however, I feel that the results are a bit trivial. If there is more rainfall and more sediment available, the debris flow will be faster and reach a longer runout. And if we separate one big event into two smaller subsequent events, the debris flow will be weaker. For practical purposes (planning etc.), it is undoubtedly useful to have such scenarios available. For an operational application, however, the*

*effort seems to be quite high and the uncertainties are also high.*

We think that the assumption "more rain and more sediment leads to a faster debris flow with longer runout" could be a hypothesis that needs to be tested, however, due to non-linearity, interactions and feedbacks and changing conditions along the flow path, in our opinion this is much too simplified for the behaviour of high mountain pro-/periglacial catchments. Moreover, scenario building is very important for scientific research proposals and practical purposes of disaster risk management. As stated above, for pro-/periglacial debris flows, comprehensible/understandable procedures for both scenario building and numerical runout modelling have so far been lacking (thus the added value of our research). Our proposed approach could be one step towards closing of this knowledge gap, but of course would need to be tested and further developed.

We are willing to focus on the doubts highlighted by reviewer 1 here and try to address them. However, "NHESS serves a wide and diverse community of research scientists, practitioners, and decision makers concerned with detection of natural hazards, monitoring and modelling, vulnerability and risk assessment, and the design and implementation of mitigation and adaptation strategies, including economical, societal, and educational aspects." Thus, the scope of the journal is different from those of, for instance, ESurf or Geomorphology. In our opinion, we do consider main challenges the addressed community is confronted with, even if the manuscript is not taking up the scientific cutting edge on research for geomorphological processes. As mentioned above, we also think that exactly this kind of paper with this practical and precisely described step-by-step procedure is often missing, bringing together real live problems with propositions for a procedure and real solutions. Nevertheless, we acknowledge the valuable comments of reviewer 1 and will strengthen the manuscript accordingly.

*Scientifically, the interplay of precipitation and sediment availability would be the most interesting aspect of debris flows. So what defines the intensity of the debris flow finally? Under which conditions is precipitation the limiting factor and under which conditions sediment availability? Unfortunately, the results presented here are restricted to a line in this 2-D parameter space since sediment availability is assumed to be a function of precipitation. As far as I can see, a single-phase flow model such as RAMMS would not be suitable for going deeper here.*

We will take up the highlighted questions to improve the quality of the manuscript, however, RAMMS is a model widely used by both scientists and practitioners. For the purposes of our manuscript, the scenario building procedures and the definition of input parameters is more important, with a focus on application to real live problems by scientists and practitioners. Thus, being aware of the limitation of each possible available model, that are the challenges we have to deal with and these models are one important basis for decision-making. And, if we cannot manage to use a simple one-phase model sensibly enough with existing data (especially for pro-

/periglacial catchments with inherently scarce to inexistent input data), how are we going to feed the more complex models (two-phase) with solid input variables? We therefore see a great need for a detailed description of an approach that has proven to deliver satisfactory results.

*In sum, I am not convinced that the manuscript in its present form provides sufficient new scientific insights, in particular in relation to its length. Sorry that I cannot be more positive at this occasion since I feel that the student's work behind was really a nice piece of work.*

We hope with the new version of the manuscript we can convince you that the manuscript is worth to be published. It's not just the scientific added value that counts, but also the important information for the practical application of models in the context of risk assessments. However, we are very happy to accept constructive criticism and strive to achieve a better balance between text length and scientific added value by focusing on the central points and, where possible, working with references to existing sources.

*Best regards,*

*Stefan Hergarten*

Many thanks Stefan, all the best and kind regards, Mauro Fischer and all co-authors

---

## Author Comment (AC2)

**Reply to RC2 (anonymous referee #2) "Comment on egusphere-2023-1190" by Mauro Fischer et al.**

In *italic*: Anonymous referee's comments

In blue normal font: our replies

*First, I would like to thank the editor Prof. Bonaccorsi for the opportunity to review this article. The study presents a multi-method approach to constructing debris flow scenarios in a small Alpine basin and performing numerical runout modeling, utilizing RAMMS-DF calibrated with the limited available data from past events. The paper introduces some compelling concepts and the proposed workflow for the study method is noteworthy. However, there are critical issues that undermine the robustness of the article.*

We thank you very much for reviewing our article. We also thank you for your appreciative remarks regarding our developed methodological concepts and workflow. We comment on and adress all the critical issues mentioned by anonymous referee #2 below.

*The main issues I have identified are as follows:*

**Length of the article:** *At 67 pages, the article is excessively long. This not only challenges reader engagement, but also suggests a potential lack of conciseness in the presentation of the research. Scientific communication typically benefits from brevity and clarity, and in this case the length may indicate superfluous details that do not contribute to the core scientific findings.*

We totally agree that our article is very long. We will shorten the manuscript and develop a clearer structure to highlight the crucial parts of the research. Actually, with <900 lines of text (including conclusions), we totally see that the main part of the manuscript is at the upper limit or above the normal manuscript length. We note that about 1/3 of the 67 pages mentioned by the reviewer belong to appendices and references…

The first submitted version of our manuscript has somewhat "guidelines" character and is therefore very long. Our rationale here was to provide researchers and practitioners a workflow with detailed description of important procedures one has to work through for debris flow hazard scenario building and numerical runout modelling for pro-/periglacial catchments with inexistent or scarce past event data. To our knowledge, for pro-/periglacial debris flows triggered by precipitation dependent events, such comprehensible/understandable procedures have so far been lacking (thus the added value of our research). We kept a lot of details in the first version of our manuscript because, from our experience, important steps are often missing or unclear in existing literature on debris flow scenario building and numerical runout

modelling in general, and do, to our knowledge, not exist yet in a comprehensive manner for pro-/periglacial catchments. Therefore, and in order to be as transparent as possible and guarantee traceability/reproducibility, we also integrated detailed and extensive appendices to the first version of the article. Nevertheless, we will shorten the manuscript wherever possible and provide references for individual working steps. Where we cannot provide references to some of the details/steps in the proposed workflow, we will leave relevant descriptions in the text (but as brief as possible).

*Software appropriateness: The use of RAMMS (Rapid Mass Movements Simulation) software is questionable in terms of its ability to support the aims set out in the article. The validation method used does not appear to support the proposed workflow. A more appropriate software package may provide a more nuanced understanding of the phenomena under investigation.*

We are totally aware that there are other tools or software packages for numerical debris flow runout modelling (see e.g., chapter 5.2 in the discussion of our manuscript). However, the aim of our paper was explicitly not to compare different tools or modelling software. For debris flow hazard assessment, RAMMS is still widely used by both scientists and practitioners. For the purpose of our manuscript, the scenario building procedures and the definition of input parameters is more important than the choice of the modelling tool. In addition, we show that we were able to reproduce the one single major debris flow event that occurred in our studied catchment sufficiently well with RAMMS. With the very limited data on past events available for our catchment, we argue that our quality assessment for the calibration of RAMMS to the studied catchment adds valuable information. It's not the methodological approach or our workflow as a whole that was validated! Testing the applicability of our workflow and providing improvements to our proposed approach would clearly need further research and is beyond the scope of our paper.
We argue that numerical modelling of our different debris flow scenarios developed in the first part of the article with RAMMS is an added value for debris flow hazard assessment in our studied catchment. Also, we critically discuss resulting simulated scenarios in chapter 5.2 of the manuscript. Being aware of the advantages and limitations of each possible available model, we show challenges one has to deal with when it comes to debris flow hazard assessment in pro-/periglacial environments with scarce past event data, and simulation results derived by RAMMS can be one important basis for decision-making. In our opinion, we do consider main challenges researchers and practitioners dealing with debris flow hazard assessments in high mountain areas are confronted with, even if the manuscript is not taking up the scientific cutting edge on research related to geomorphological processes or more sophisticated numerical modelling. If we cannot manage to use a simple one-phase model sensibly enough with existing data (especially for pro/periglacial catchments with inherently scarce to inexistent input data), how are we going to feed the more complex models (two-phase) with solid input variables? We therefore think that our work proposes important concepts and an approach that has proven to deliver satisfactory results.

*Literature review: The literature review presented does not meet the standards required for a study of this nature. There are more sophisticated flow models available, such as SPH (Smoothed Particle Hydrodynamics) cited in the work of Pastor et al. from the Universidad Politécnica de Madrid on SPH Geoflow, which could provide deeper insights into aspects not adequately considered in this paper. For example, the potential consolidation of material during movement and the erosion of material that joins the mass during runout are important factors that appear to have been overlooked or inadequately considered.*

In our opinion, our referenced literature is appropriate and very comprehensive. With all respect, all co-authors together bring in decades of scientific and practical experience when it comes to high mountain geomorphology, cryospheric and torrential processes and debris flow hazard assessment. We would be willing to add more references to other flow models, but again: For the purposes of our manuscript, the scenario building procedures and the definition of input parameters is more important than the choice of the modelling tool. Being aware of the limitations of RAMMS, we show challenges one has to deal with when modelling debris flow scenarios in pro-/periglacial environments with scarce past event data, and simulation results derived by RAMMS, a tool still widely used today by practitioners and scientists, are one important basis for decision-making/hazard assessment. Again, if we cannot manage to use a simple one-phase model sensibly enough with existing data (especially for pro/periglacial catchments with inherently scarce to inexistent input data), how are we going to feed the more complex models (two-phase) with solid input variables? We argue that for the special case of debris flow scenario building in pro-/periglacial catchments with scarce past event data and inherent difficulties related to input variable definition for numerical modelling, it's not guaranteed at all that more sophisticated models than RAMMS would produce "better" results. For catchments with no or scarce past event data, and especially for highly dynamic and imbalanced pro-/periglacial catchments, a "validation" of derived scenarios and numerical modelling results ex ante is per se impossible. As we write in the discussion: "even if the different work steps comprised in scenario building and runout modelling (Fig. 4) of PPDFs are carried out as seriously as possible by experts, simulation results of PPDF scenarios should be handled with care and treated as what they are: best possible representations of what could possibly happen in the future." We totally agree with anonymous referee #2 that erosion along the flow path can be very important. Even though it could be considered with RAMMS, we did not account for erosion along the flow path based on our field surveys which showed that entrainment along the channel would be minor if not negligible compared to loose material released in the debris flow initiation zone (see section 5.2 and Appendix A).

*Given the current form and assumptions of the paper, and unless there is a significant reduction in length and an increase in methodological rigour, I regret to say that the paper does not meet the criteria for publication. While the authors' efforts are appreciated, the paper should be rejected in its current form. In order to make a significant contribution to the field, it is essential that the paper is refined to*

*succinctly communicate the research, employ appropriate validation methods, and thoroughly engage with the existing literature.*

We are willing to shorten and restructure the paper, and hope with the new version of the manuscript we can convince you that the manuscript is worth to be published. We are very happy to accept constructive criticism and strive to achieve a better balance between text length and scientific added value. With all respect, but considering the expertise in high mountain geomorphodynamics and debris flow hazard assessment of all co-authors together, we dare to say that we are convinced that our work is a significant contribution to the field, even though it may not be at the scientific cutting edge when it comes to increase geomorphological process understanding and more sophisticated numerical debris flow modelling (which was also clearly not a main goal of our work!). For hazard assessment in rapidly changing high mountain catchments, we provide comprehensible/understandable procedures for scenario building and runout modelling of pro-/periglacial debris flows triggered by precipitation dependent events, something that, to our knowledge, has clearly been lacking in literature so far. We provide a methodological approach to deal with real life problems that might also be used by practitioners for hazard assessment (what should be done when assessing debris flow hazards from pro-/periglacial initiation zones with clearly given disposition for debris flows but with no or only scarce data on past events). We did not validate our proposed workflow per se (but only commented on the quality of the calibration of RAMMS to our studied catchment)! Testing the applicability of our workflow and providing improvements to our proposed approach would clearly need further research and is beyond the scope of our paper.

We want to thank anonymous referee #2 again for taking time to review our manuscript and for the constructive and important feedbacks. Kind regards, Mauro Fischer and all co-authors